# BROWSECOMP-ZH: BENCHMARKING WEB BROWSING ABILITY OF LARGE LANGUAGE MODELS IN CHINESE

## ABSTRACT

As large language models (LLMs) evolve into web-interacting agents, their ability to retrieve and reason over real-time information has become a crucial benchmark for general intelligence. However, existing benchmarks such as BrowseComp focus solely on English, neglecting the linguistic, infrastructural, and retrieval-specific challenges posed by other information ecosystems—particularly the Chinese web. We present **BrowseComp-ZH**, a high-difficulty, natively-constructed benchmark designed to assess LLM agents' web browsing abilities in Chinese. Rather than translating from English, all questions in BrowseComp-ZH are written from scratch by native speakers to reflect authentic information-seeking behaviors and cultural contexts. The dataset comprises 289 multi-hop questions across 11 diverse domains, each reverse-engineered from a short, verifiable answer and filtered through a two-stage quality control pipeline to ensure retrieval hardness and answer uniqueness. We evaluate over 20 leading LLMs and agentic search systems. Despite strong language and retrieval abilities, most models perform poorly: many score below 10% accuracy, and only a few exceed 20%. Even the best system achieves just 42.9% accuracy. These results highlight the considerable difficulty of BrowseComp-ZH, where success requires not only robust retrieval strategies but also advanced multi-hop reasoning and information reconciliation—abilities that remain challenging for current models. BrowseComp-ZH thus serves as a stress test for web-interactive LLMs beyond English, offering a rigorous and linguistically diverse evaluation framework to guide future research on multilingual agent capabilities.

## 1 INTRODUCTION

As large language models (LLMs) evolve from static knowledge repositories to dynamic agents capable of using external tools, tasks that involve web browsing have emerged as a critical lens through which to evaluate their real-world reasoning and information-seeking capabilities (Xi et al., 2025). By interacting with search engines and navigating live web content, LLMs can augment their internal knowledge with up-to-date external evidence, retrieve context-specific information, and perform multi-hop reasoning across heterogeneous sources. This browsing ability extends the temporal scope of LLMs while enabling them to tackle questions that lie beyond the reach of pretraining, such as time-sensitive facts or obscure entity relations that require targeted retrieval.

While an increasing number of studies (Li et al., 2023b; Fernández-Pichel et al., 2024; Fan et al., 2024; Vu et al., 2023; Lai et al., 2025; He et al., 2024; Lai et al., 2024; Xiong et al., 2024) demonstrate that web browsing greatly improves LLM performance on downstream tasks, there remains a surprising lack of direct evaluation of browsing capabilities themselves—i.e., the ability of LLMs to effectively retrieve, filter, and reason over information from the web. This evaluation is crucial for assessing the true web-browsing competence of LLMs and understanding their potential to tackle real-world tasks that require dynamic information retrieval. To address this gap, Wei et al. (2025) introduced BrowseComp, a browsing competition benchmark dataset that challenges English-language agents to search and reason over difficult-to-access information.

However, existing benchmarks like BrowseComp focus exclusively on the English-language web, leaving open the question of how LLM agents perform in other major information ecosystems—most notably, the Chinese web. A seemingly straightforward solution is to translate English benchmarks into Chinese. Yet this approach fundamentally fails to capture the depth and complexity of native

| | |
|---|---|
| 话题：艺术 **Topic**：Art | 话题：影视 **Topic**：Film & TV |
| 问题：在中国传统艺术中，有一种特殊的绘画形式，起源于元代，盛行于清末，传说是由一位古代知名画家在酒后兴起所创。这种艺术形式在2010~2015年之间被列入某省级非物质文化遗产名录。绘制这种艺术形式需要画家精通各种画法，并且要善书各种字体。请问这种艺术形式是什么？ | 问题：某知名电视剧，女二号（演员）在1993年进入演艺圈。女一号（演员）的现任丈夫是浙江湖州人。男一号（演员）6年后登上了春晚舞台。问该电视剧是什么？ |
| **Question:** In traditional Chinese art, there is a unique form of painting that originated in the Yuan Dynasty and became popular during the late Qing Dynasty. It is said to have been created by a famous ancient painter who was inspired by alcohol. Between 2010 and 2015, this art form was included in the provincial intangible cultural heritage list of a certain region. To paint in this style, artists must be proficient in various painting techniques and skilled in writing different types of calligraphy. What is this art form called? | **Question:** In a well-known TV drama, the second female lead (actress) entered the entertainment industry in 1993. The current husband of the first female lead (actress) is from Huzhou, Zhejiang. The first male lead (actor) performed on the CCTV Spring Festival Gala six years later. What is the name of this TV drama? |
| 答案：锦灰堆  **Answer:** Heaps of Brocade and Ash | 答案：父母爱情  **Answer:** Love of Parents |

Figure 1: Two data samples from BrowseComp-ZH with their English translations.

Chinese usage for three reasons. First, translated questions—regardless of quality—tend to retain syntactic and lexical artifacts from the source language (a phenomenon known as translationese), leading to unnatural expressions that do not reflect authentic Chinese linguistic features such as dropped subjects, classifiers, or idiomatic phrasings. These artifacts are well-documented in multilingual NLP (Kong & Macken, 2025; Enomoto et al., 2025), and native datasets like OCNLI (Hu et al., 2020) avoid them by constructing prompts directly in Chinese. Second, language is deeply embedded in cultural context, and many semantically rich or emotionally nuanced expressions in Chinese—such as implicit praise, historical allusions, or honorific forms—are easily flattened or distorted in translation. This loss of cultural grounding can lead to biased or misleading assessments of model capabilities (Liu, 2025). Finally, evaluating agents on the Chinese web presents retrieval challenges that are absent in English, including fragmented indexing across Baidu, WeChat, Zhihu, and Little Red Book; censorship-related link rot; and homophones. Translated benchmarks often fail to expose models to these retrieval dynamics, as they default to English-centric search pathways. Collectively, these factors underscore that only natively constructed benchmarks can provide realistic and rigorous evaluation of Chinese-language agents.

To address this gap, we introduce **BrowseComp-ZH**, the first benchmark specifically designed to evaluate the web browsing and reasoning abilities of LLMs within the Chinese information ecosystem. BrowseComp-ZH consists of 289[*] expert-authored questions spanning 11 diverse domains—including film, technology, law, and medicine. Figure 1 presents two representative examples from BrowseComp-ZH. Each question features multiple constraints and a single verifiable answer, and is carefully designed to require multi-hop reasoning and the integration of fragmented information scattered across Chinese web platforms. These characteristics make the benchmark resistant to shallow keyword matching and ideal for assessing deep retrieval and reasoning abilities.

Compared to existing benchmarks, BrowseComp-ZH offers three key advantages: (1) **Stricter construction pipeline.** All examples are created in native Chinese by expert annotators (each holding at least a master's degree and relevant domain knowledge). The process starts from a known factual answer, from which annotators reverse-engineer a question with multiple constraints that is *difficult to retrieve yet easy to verify*. Each sample must pass a triple-engine test—-Baidu, Bing, and Google—and is retained only if the correct answer does not appear on the first page of any search engine. (2) **Answer uniqueness auditing.** We implement a two-stage validation process. In Stage 1, annotators perform time-constrained searches to confirm unretrievability. In

---

[*]Following the design philosophy of stress-test-style benchmarks such as HumanEval Li & Murr (2024), WSC-273 Levesque et al. (2012), and MMSearch Jiang et al. (2024), we prioritize depth and difficulty over size. As shown in our experiments, this scale is sufficient to induce meaningful performance differences across models. For further discussion, see Appendix A.

Stage 2, we deploy multiple top-tier agents to generate plausible answers, which are then manually vetted. Any sample with multiple valid answers is removed to ensure that all questions have a single, unambiguous solution. (3) **Broader system evaluation.** Unlike BrowseComp, which evaluates a small number of English-only agents, BrowseComp-ZH benchmarks **over 20 systems**, including open-source models (e.g., DeepSeek-R1 (Guo et al., 2025), Qwen2.5-72B-Instruct (Yang et al., 2024)), closed-source models (e.g., GPT-4o (OpenAI, 2024a), Claude-3.7 (Anthropic, 2025), Gemini-2.5-Pro (Google, 2024b)), and commercial agentic search products (e.g., DeepResearch (OpenAI, 2025b), Doubao (ByteDance, 2025), Perplexity (Perplexity, 2025)). This broader system coverage enables more comprehensive diagnosis of LLM agent capabilities under realistic Chinese-language search conditions.

Our evaluation offers a comprehensive analysis of model capabilities across system types, leading to several key findings:

1. Naive large language models, irrespective of parameter scale or training data size, exhibit consistently poor performance on the benchmark, with accuracies often remaining below 10%. Representative examples include Qwen2.5-72B-Instruct, Llama 4, and GPT-4o.

2. Models endowed with inherent reasoning capabilities exhibit consistent improvements in accuracy. For example, DeepSeek-R1 surpasses DeepSeek-V3 by 13.6%, while Claude-3.7-Sonnet outperforms Claude-3.5 by 10.4%.

3. Agentic systems augmented with external retrieval mechanisms tend to achieve higher scores. Notably, OpenAI's DeepResearch and Doubao (Deep Search) demonstrate that well-orchestrated retrieval and reasoning pipelines can substantially enhance performance, achieving accuracies of 42.9% and 26.0%, respectively.

4. While well-designed retrieval pipelines can significantly improve performance, not all systems benefit from retrieval integration. For instance, enabling web search for DeepSeek-R1 results in a substantial decline in performance, with accuracy dropping from 22.5% in the direct-answer (no web access) setting to 7.6% when web search is enabled.

5. Human performance sets a realistic yet non-trivial baseline. In our human evaluation, 29 strong searchers achieved an average accuracy of 17.6%—outperforming most open-source and closed-source models, and even matching or exceeding many commercial search agents. This underscores the difficulty of the benchmark and its utility in diagnosing model shortcomings beyond synthetic or over-optimized datasets.

## 2 RELATED WORK

With the increasing ability of large language models (LLMs) to use external tools, recent research has focused on evaluating their capacity to retrieve and reason over real-world information. Representative works such as WebGPT (Nakano et al., 2021), Toolformer (Schick et al., 2023), and ReAct (Yao et al., 2023) explore how LLMs leverage search engines, tool usage, and reasoning strategies to tackle complex question-answering tasks. In parallel, retrieval-augmented generation (RAG) frameworks (Guu et al., 2020; Lewis et al., 2020) have been widely adopted in QA (Wiratunga et al., 2024), summarization (Edge et al., 2024), and fact verification (Martin et al., 2024), serving as a mechanism for injecting external knowledge into LLMs.

To assess retrieval capabilities, a variety of widely used English benchmarks have been proposed, including TriviaQA (Joshi et al., 2017), HotpotQA (Yang et al., 2018), FEVER (Thorne et al., 2018), KILT (Petroni et al., 2020), and GAIA (Mialon et al., 2023). These datasets cover multi-hop reasoning, knowledge-intensive QA, and fact checking, typically relying on structured sources like Wikipedia and StackExchange. However, since many answers can be retrieved via simple keyword searches, these benchmarks often fail to evaluate an agent's ability to plan complex search trajectories and synthesize information across documents. Wei et al. (2025) addresses this by reverse-designing queries from known answers with multiple retrieval constraints, requiring multi-hop search and cross-page reasoning. While it offers finer-grained evaluation for web browsing agents, it remains confined to the English web and lacks generalizability to non-English environments with fragmented platforms and diverse linguistic structures.

However, extending such evaluations to Chinese poses unique challenges. Several retrieval-related datasets have emerged for the Chinese web, but each presents notable limitations. Hu et al. (2024)

evaluates Chinese web search agents but lacks rigorous control over task difficulty and answer accessibility. Xu et al. (2024) focuses on dynamic, time-sensitive QA tasks but does not emphasize multi-hop retrieval. Lyu et al. (2025) evaluates RAG systems under the CRUD (Create, Read, Update, Delete) paradigm, yet primarily emphasizes generation quality over retrieval path validation. Liu et al. (2023) and Li et al. (2023a) focus on domain-specific medical QA, but do not evaluate open-ended retrieval or browsing strategies.

To address these limitations in Chinese retrieval benchmarking, BrowseComp-ZH introduces three key innovations: (1) reverse-designed tasks in native Chinese to avoid translation artifacts; (2) rigorous multi-step validation to ensure high retrieval difficulty and answer verifiability; and (3) broad model coverage for evaluating both open-source and proprietary agents. It establishes a high-difficulty benchmark for Chinese web retrieval, supporting systematic evaluation of agents' ability to navigate fragmented, unstructured, and linguistically diverse Chinese information sources.

## 3   THE BROWSECOMP-ZH DATASET

### 3.1   DATASET CONSTRUCTION

Following Wei et al. (2025), we adopt a reverse dataset construction strategy, starting from a factual answer and crafting an elaborate, multi-constraint question to make direct retrieval non-trivial. To ensure high-quality annotation, we recruited 10 expert annotators who are native Chinese speakers and hold Master's or PhD degrees, with extensive experience in both LLM usage and web search. The overall process comprises the following two stages:

**Stage 1: Topic and Answer Selection**   Each annotator selects at least 5 topics from a predefined list spanning *Film & TV*, *Technology*, *Art*, *History*, *Sports*, *Music*, *Geography*, *Policy & Law*, *Medicine*, *Video Games*, and *Academic Research*, based on their personal interests. For each topic, they identify several factual answers (e.g., person names, dates, titles, institutions) that meet two criteria: (1) The selected facts must be objective statements that can be independently verified through reliable sources, without the need for interpretation or inference; and (2) they must be concrete and specific enough to exclude overly generic or widely known common-sense facts.

**Stage 2: Reverse Question Design**   Building on the selected factual answers, annotators construct complex questions that require the integration of contextual cues and external knowledge. This stage adheres to two core principles. First, each question follows a *multi-constraint design*, incorporating at least two types of conditions such as temporal, spatial, categorical, or descriptive constraints. Intuitively, the greater the diversity and specificity of the constraints, the smaller the set of possible answers, thereby increasing the likelihood that only the intended answer satisfies all conditions. Second, to ensure *non-trivial retrieval*, annotators are required to perform two rounds of self-verification on each constructed question. In the first round, they test the question on Baidu, Bing, and Google using three distinct keyword combinations per search engine. If the correct answer appears on the first page of any search engine, the question is deemed too easy and must be revised. In the second round, the question is evaluated using GPT-4o and DeepSeek, both equipped with web-enabled search capabilities. If both models consistently retrieve the correct answer with minimal effort, the question is further refined—for example, by introducing implicit constraints or obfuscating key lexical cues—to increase its complexity. This process yields 480 preliminary samples, which are subsequently reviewed through the quality control procedure described in Section 3.2.

### 3.2   QUALITY CONTROL

To ensure the rigor and challenge of BrowseComp-ZH, we implement a two-stage quality control protocol: one focusing on question difficulty and the other on answer uniqueness.

**Stage 1: Question Difficulty Validation**   Although a self-verification step is included during data construction, differences in annotators' domain knowledge and search proficiency may lead to inconsistent difficulty assessments. For example, an annotator might use overly simple search keywords, resulting in a question being mistakenly classified as non-trivially retrievable. To address this, we introduce a cross-checking phase in which each annotator validates questions written by

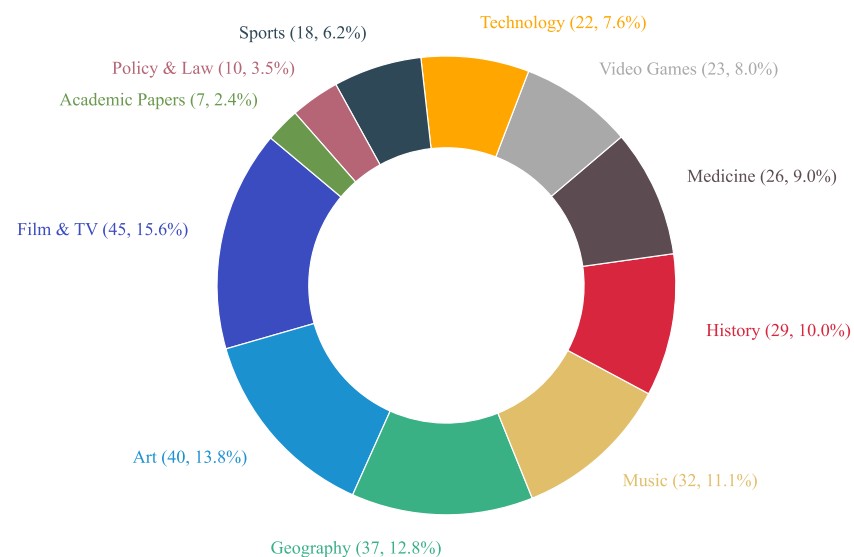

Figure 2: Distribution of samples across 11 topic domains in BrowseComp-ZH dataset.

others using only search engines, without assistance from LLMs. A strict 10-minute time limit is imposed per question. If the correct answer is found within the time limit, the question is labeled as *low difficulty*; otherwise, if the answer is not found and the question structure is deemed logical and verifiable, it is labeled as *high difficulty*. Through this process, annotators identified 76 low-difficulty questions, which were subsequently removed, leaving 404 high-difficulty candidates for the final dataset.

**Stage 2: Answer Uniqueness Validation** This stage aims to ensure, as much as possible, that each question admits only one correct and unambiguous answer that satisfies all specified constraints. To this end, we adopt a human-in-the-loop procedure involving multiple top-performing AI agents—namely OpenAI DeepSearch, Perplexity, Doubao (with deep reasoning), OpenAI O1, and Gemini 2.5-Pro—selected based on their demonstrated ability to retrieve accurate information and perform complex reasoning. These agents are prompted to generate both the reasoning process and the final answer for each of the 404 high-difficulty candidate questions. Annotators then manually review the outputs to determine whether any alternative answer, differing from the original one, also satisfies all the multiple constraints designed in the question, such as temporal, spatial, or categorical conditions. If such alternative answers are identified, the corresponding question is deemed ambiguous and discarded. This process filters out 115 ambiguous samples, resulting in a final benchmark of 289 validated questions that maintain high levels of difficulty and answer uniqueness[2].

### 3.3 DATA STATISTICS

In this section, we present statistics on the topic distribution, question and answer length distribution of our curated BrowseComp-ZH dataset.

**Topic distribution.** Fig. 2 presents the distribution of samples across 11 topic domains in the BrowseComp-ZH dataset. As illustrated, the most represented categories include *Film & TV* (15.6%), *Art* (13.8%), and *Geography* (12.8%), reflecting the diverse interests of annotators and a broad coverage of Chinese web content. The dataset also features *Music* (11.1%), *History*(10.0%), and *Medicine* (9.0%). Conversely, topics like *Policy & Law* (3.5%) and *Academic Papers* (2.4%) have fewer samples, likely due to the complexity of sourcing factual answers from these areas. The distribution underscores the multi-disciplinary nature of the dataset, aimed at evaluating language models across a wide array of knowledge domains.

---

[2]While this procedure substantially improves the likelihood of answer uniqueness, we acknowledge that perfect uniqueness cannot be fully guaranteed; a detailed discussion of this limitation is provided in Section 5.

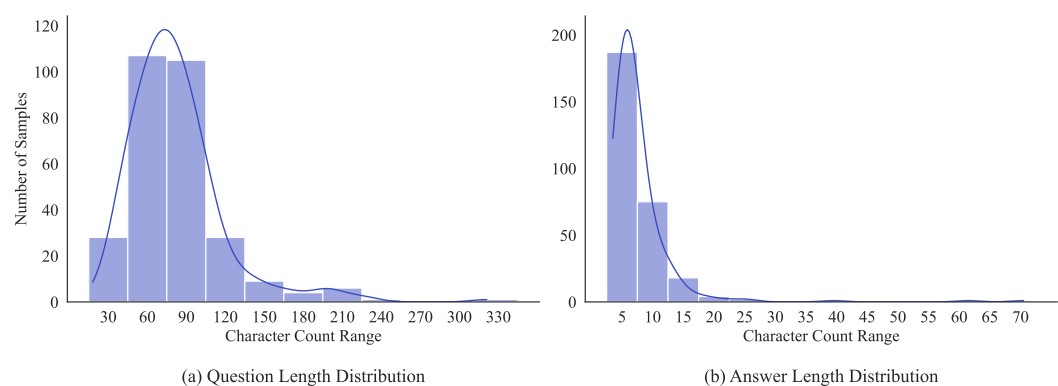

(a) Question Length Distribution          (b) Answer Length Distribution

Figure 3: Distribution of question and answer lengths in the BrowseComp-ZH dataset.

**QA Length distribution.** Fig. 3 illustrates the distribution of question and answer lengths in the BrowseComp-ZH dataset. As shown in Fig. 3 (a), the question length predominantly ranges between 60 to 90 characters, with the majority of questions falling within this interval. This suggests that questions are designed to be succinct but information-dense, requiring substantial detail to challenge the models. In Fig. 3 (b), the answer length distribution shows that answers are generally short, typically falling within the range of 5 to 10 characters. This aligns with the design principle of providing precise and verifiable answers, ensuring that the responses are direct and concise. These length distributions reflect the high information density of the dataset, emphasizing both complexity in the questions and simplicity in the answers.

## 4 BENCHMARKS

### 4.1 EVALUATED SYSTEMS

We evaluate a wide range of state-of-the-art open-source and proprietary models, as well as mainstream AI search products on BrowseComp-ZH, aiming to provide a diverse, comprehensive, and insightful benchmark.

- **Open-source models:** DeepSeek-V3 (Liu et al., 2024), DeepSeek-R1 (Guo et al., 2025), Qwen2.5-72B-Instruct (Yang et al., 2024), Qwen3-235B-A22B (Yang et al., 2024), QwQ-32B (Qwen_Team, 2024), and Llama4 (maverick-instruct-basic) (Meta, 2024).

- **Closed-source models:** GPT-4o (OpenAI, 2024a), O1 (OpenAI, 2024b), O4-mini (OpenAI, 2025a), Claude-3.5-Sonnet (20240620 ) (Anthropic, 2024), Claude-3.7-Sonnet (20250219) (Anthropic, 2025), Gemini-2.0-Flash (Google, 2024a), Gemini-2.5-Pro (preview-03-25) (Google, 2024b), and Qwen2.5-MAX (qwen-max-2025-01-25) (Qwen_Team, 2025)

- **AI search products:** DeepResearch (OpenAI, 2025b), Grok-3 (xAI, 2025), Perplexity (Research mode) (Perplexity, 2025), Doubao (with both deep search and standard modes) (ByteDance, 2025), Kimi (deep think version) (MoonShot_AI, 2025), Yuanbao (Hunyuan Model) (Tencent, 2025), and DeepSeek (with both deep think and standard modes) (DeepSeek, 2025).

### 4.2 EVALUATION SETUP

**Open-source and Closed-source Models.** For O1 and Claude-3.7, inference was performed using default decoding parameters, as temperature and top-p settings are not user-configurable. For all other open-source and closed-source models, we followed the decoding configuration used in DeepSeek R1 (Guo et al., 2025), setting the temperature to 0.6 and top-p to 0.95. All evaluations were conducted via API calls, with queries programmatically submitted and responses automatically collected. Final answers were extracted from the returned text using regular expressions, based on a predefined output format instructed to the models. The extracted answers were then graded using GPT-4o, following prompts adapted from (Wei et al., 2025). Full prompt details are included in Appendix Fig. 5. To

capture performance variability, we conducted three independent runs for each model and reported the mean and standard deviation of evaluation metrics.

**AI Search Products.** Due to the lack of accessible APIs, AI search products were evaluated via manual GUI-based interactions, where human annotators input each query and recorded the returned answers. These systems typically undergo additional post-training to align with product-specific use cases, which often reduces their ability to follow instructions precisely. As a result, regular-expression-based answer extraction was no longer applicable, and annotators need to manually extract final answers and verify their consistency with the ground truth. Given the high labor cost of this process, the evaluation was limited to a single run per sample. The full instructions provided to both model types are included in Appendix Fig. 4.

**Evaluation Metrics.** To assess model performance, we report both accuracy and expected calibration error (ECE). Accuracy reflects whether the model's prediction is correct, while ECE captures how well the model's predicted confidence aligns with its actual correctness. This provides insight into whether the model is over- or under-confident—particularly important in retrieval-based or high-stakes applications where confidence reliability matters. To compute ECE, we partition the predicted confidence scores into five bins: [0–0.2), [0.2–0.4), [0.4–0.6), [0.6–0.8), and [0.8–1.0]. For each bin, we calculate the absolute difference between the average predicted confidence and the empirical accuracy. A weighted average is then computed across all bins to yield the final ECE:

$$\text{ECE} = \sum_{i=1}^{B} \frac{n_i}{N} |acc(i) - conf(i)|,$$

where $B$ is the number of bins, $n_i$ is the number of samples in the $i$-th bin, $acc(i)$ is the empirical accuracy in that bin, $conf(i)$ is the average predicted confidence, and $N$ is the total number of samples.

**Human Evaluation.** To establish a reference point for human-level performance, we conducted a dedicated human evaluation involving 29 strong searchers from diverse educational backgrounds, who were asked to solve BrowseComp-ZH questions using only search engines, without access to any AI tools. To ensure feasibility while preserving challenge, each question had a strict time limit of 30 minutes.

## 4.3 RESULTS AND ANALYSIS

**Overall Performance** As shown in the performance comparison of models and AI search products on our benchmark (Tab. 1), several systems, such as Qwen2.5-72B-Instruct (6.8% accuracy), GPT-4o (7.0%), and Claude-3.5-Sonnet (5.8%), exhibited limited performance, consistently achieving low accuracy scores across tasks, highlighting the challenging nature of the dataset we proposed. Among all evaluated systems, OpenAI's DeepResearch achieved the highest accuracy (42.9%), followed by OpenAI's O1 (28.9%) and Google's Gemini-2.5-Pro (28.1%). Interestingly, even models without browsing capabilities, such as DeepSeek R1, O1, and Gemini-2.5-Pro, which rely solely on their internal world knowledge, managed to achieve accuracies exceeding 20%, demonstrating the strength of high-capacity language models. Overall, AI search products equipped with retrieval mechanisms outperformed other categories, followed by closed-source models, while open-source models performed the worst on this challenging benchmark.

**Analysis of Reasoning Ability of LLM** Since answering the complex questions in our proposed BrowseComp-ZH benchmark requires both extensive world knowledge and strong reasoning abilities, we further investigate the impact of reasoning on model performance. As summarized in Tab. 1, we compare models with and without explicit reasoning mechanisms, including DeepSeek-V3 versus DeepSeek-R1, Claude-3.5-Sonnet versus Claude-3.7-Sonnet, and Gemini-2.0-Flash versus Gemini-2.5-Pro. Across all comparisons, models enhanced with reasoning capabilities consistently demonstrate substantial performance gains. For example, DeepSeek-R1 achieves an accuracy of 22.5%, markedly improving upon DeepSeek-V3's 8.9%, while Claude-3.7-Sonnet outperforms Claude-3.5-Sonnet (16.2% vs. 5.8%). A similar pattern emerges among AI search products. Doubao (Deep Search) achieves a nearly 8% absolute improvement over Doubao (Standard) (26.0% vs.

| Model | Reasoning | Browsing | Accuracy (%) | Calibration (%) | Enterprise |
|---|---|---|---|---|---|
| Open-Source Models | | | | | |
| DeepSeek-V3 | ✗ | ✗ | 8.9 ± 0.2 | 71.4 ± 0.4 | DeepSeek |
| DeepSeek-R1 | ✓ | ✗ | 22.5 ± 1.2 | 60.1 ± 1.0 | DeepSeek |
| Qwen2.5-72B-Instruct | ✗ | ✗ | 6.8 ± 0.3 | 62.7 ± 1.2 | Alibaba |
| Qwen3-235B-A22B (Non-Thinking) | ✗ | ✗ | 6.8 ± 1.0 | 82.3 ± 1.6 | Alibaba |
| Qwen3-235B-A22B (Thinking) | ✓ | ✗ | 14.3 ± 1.0 | 66.2 ± 1.2 | Alibaba |
| QwQ-32B | ✓ | ✗ | 9.3 ± 1.2 | 66.8 ± 1.6 | Alibaba |
| Llama4 | ✗ | ✗ | 6.2 ± 1.3 | 67.0 ± 2.3 | Meta |
| Closed-Source Models | | | | | |
| GPT4o | ✗ | ✗ | 7.0 ± 0.7 | 71.9 ± 0.8 | OpenAI |
| O1 | ✓ | ✗ | 28.9 ± 0.2 | 53.5 ± 1.0 | OpenAI |
| O4-mini | ✓ | ✗ | 15.7 ± 0.3 | 40.6 ± 0.8 | OpenAI |
| Claude-3.5-Sonnet | ✗ | ✗ | 5.8 ± 0.6 | 76.5 ± 0.8 | Anthropic |
| Claude-3.7-Sonnet | ✓ | ✗ | 16.2 ± 1.2 | 71.2 ± 0.7 | Anthropic |
| Gemini-2.0-Flash | ✗ | ✗ | 7.6 ± 0.7 | 73.4 ± 0.3 | Google |
| Gemini-2.5-Pro | ✓ | ✗ | 28.1 ± 0.9 | 59.8 ± 0.4 | Google |
| Qwen2.5-MAX | ✗ | ✗ | 7.4 ± 0.9 | 78.5 ± 0.5 | Alibaba |
| AI Search Products | | | | | |
| OpenAI DeepResearch | ✓ | ✓ | **42.9** | 9 | OpenAI |
| Grok3 (Research) | ✓ | ✓ | 12.9 | 39 | xAI |
| Perplexity (Research) | ✓ | ✓ | 22.6 | 53 | Perplexity |
| Doubao (Deep Search) | ✓ | ✓ | 26.0 | 61 | ByteDance |
| Doubao (Standard) | ✗ | ✓ | 18.7 | 37 | ByteDance |
| Kimi (Deep Think) | ✓ | ✓ | 8.0 | 58 | Moonshot |
| Yuanbao (Hunyuan) | ✓ | ✓ | 12.2 | 56 | Tencent |
| DeepSeek (Deep Think) | ✓ | ✓ | 7.6 | 65 | DeepSeek |
| DeepSeek (Standard) | ✗ | ✓ | 4.8 | 66 | DeepSeek |
| Human | ✓ | ✓ | 17.6 | - | - |

Table 1: Performance of various models and AI search products on the BrowseComp-ZH benchmark. For open-source and closed-source models, we report the mean ± standard deviation over three runs. **Bold** indicates the best performance across all models, and underline denotes the second-best.

18.7%), highlighting the benefits of enhanced reasoning in facilitating more accurate and iterative retrieval for complex queries. However, this trend is less evident in DeepSeek's AI search products. Notably, all versions of DeepSeek's search system perform only a single round of retrieval, limiting the extent to which improved reasoning capabilities can be leveraged. Consequently, DeepSeek's Deep Search variant does not exhibit a significant performance advantage over its standard counterpart.

**Analysis of AI Search Products** Currently, AI search systems can be broadly categorized into two types: those that perform a single retrieval to answer a user's query (e.g., Kimi, Tencent Yuanbao based on the Hunyuan model, and DeepSeek), and those that conduct multiple rounds of retrieval, iteratively refining or expanding the search based on the query and intermediate results (e.g., DeepResearch, Perplexity, and Doubao). Statistical analysis shows that systems employing multi-round retrieval achieve significantly higher accuracy, with DeepResearch reaching 42.9%, Doubao (Deep Search) achieving 26.0%, and Perplexity attaining 22.6%, compared to single-retrieval systems such as Kimi (8.0%), Yuanbao (12.2%), and DeepSeek (7.6%). This trend aligns with the nature of tasks in BrowseComp-ZH, which often involve multi-faceted queries, making it challenging to obtain accurate answers through a single retrieval operation.

Notably, we observe a counterintuitive phenomenon: enabling search functionality for DeepSeek-R1 leads to a substantial decline in performance, with accuracy dropping from 22.5% in the direct-answer (no web access) setting to 7.6% when web search is enabled. We hypothesize that this degradation arises because, without effective alignment mechanisms, the model may rely on less reliable retrieved content, which in turn overrides its more accurate internal knowledge. This observation highlights a critical challenge for large language models: effectively reconciling retrieved evidence with internal representations remains non-trivial. Furthermore, integrating retrieval capabilities without robust

post-retrieval reasoning and alignment strategies may, in some cases, hinder rather than enhance model performance.

**Comparison with Human Performance.** The average accuracy achieved by human participants was 17.6%, which serves as a meaningful baseline for interpreting model performance. Notably, the vast majority of open-source and closed-source models—including Qwen2.5-72B, GPT-4o, Claude-3.5-Sonnet, and even Claude-3.7-Sonnet—fail to surpass human performance. This underscores the difficulty of BrowseComp-ZH, which requires deep reasoning and precise information synthesis across fragmented Chinese web content. Only a handful of models, such as DeepSeek-R1 (22.5%), O1 (28.9%), and Gemini-2.5-Pro (28.1%), manage to outperform humans, largely benefiting from advanced reasoning capabilities. More prominently, AI search products equipped with multi-round retrieval and reasoning mechanisms—such as DeepResearch (42.9%) and Doubao (26.0%)—consistently exceed human-level accuracy, suggesting that tight integration between browsing and reasoning can unlock superior performance. These results validate BrowseComp-ZH as a challenging yet discriminative benchmark that meaningfully differentiates retrieval-free models, agentic systems, and human searchers.

**Analysis of Calibration Error** We also evaluate the model's calibration error, which measures the alignment between predicted confidence scores and actual accuracy. Following the methodology adopted in Wei et al. (2025), we require models to provide confidence estimates alongside their predictions during evaluation. As shown in Tab. 1, integrating search functionality results in increased calibration errors. For instance, the calibration error for DeepSeek-R1 increases from 60.1% in the direct-answer setting to 65% when search is enabled. This trend is consistent with the observations reported in BrowseComp.

**Case Study** Due to space limitations in the main text, we include a detailed case study of four leading AI systems in Appendix E.

## 5    CONCLUSION AND LIMITATIONS

In this work, we introduce **BrowseComp-ZH**, the first benchmark specifically designed to evaluate the web browsing and reasoning capabilities of LLMs within the Chinese information ecosystem. It consists of 289 high-quality, human-authored questions across 11 diverse domains, each requiring multi-hop retrieval, information filtering, and logical reasoning to arrive at a concise and verifiable answer. To ensure difficulty and answer uniqueness, we implement a rigorous two-stage quality control process involving multi-engine validation and human-in-the-loop verification. We evaluate over 20 representative systems, covering both open- and closed-source LLMs as well as commercial AI search products. Results show that most standalone LLMs (e.g., GPT-4o, Qwen2.5-72B, Llama4) struggle with the benchmark, achieving less than 10% accuracy. Models with stronger reasoning capabilities (e.g., O1, Gemini-2.5-Pro) perform notably better, and agentic systems with multi-turn retrieval pipelines (e.g., DeepResearch, Doubao Deep Search) achieve the highest scores—demonstrating the value of integrating retrieval with iterative reasoning. Additionally, we conduct a dedicated human evaluation with 29 strong searchers, who achieve an average accuracy of 17.6%, outperforming most models and even rivaling several AI search agents. This highlights both the difficulty and the diagnostic utility of BrowseComp-ZH, offering a realistic and high-resolution benchmark for assessing next-generation web-browsing agents in Chinese.

**Limitations** Despite its rigorous design, **BrowseComp-ZH** has several limitations. First, compared to its English counterpart BrowseComp, the dataset size is relatively smaller due to the high labor cost of curating Chinese queries with guaranteed answer uniqueness and multi-stage human verification. However, as shown in our experiments, this compact design still induces meaningful performance differences across systems and effectively captures model deficiencies in browsing and reasoning. For a detailed discussion, see Appendix A. Second, the guarantee of answer uniqueness is inherently limited by the reverse-design paradigm, as some queries may admit multiple plausible answers despite extensive quality control. We address this through multi-agent validation and human adjudication, and will further refine constraint formulations to minimize ambiguity.

**Ethics & Societal Impacts**   The dataset was constructed via a transparent, reproducible process: expert annotators reverse-engineered questions from verifiable answers using multi-constraint design, with a two-stage human-in-the-loop validation to ensure difficulty and answer uniqueness. To mitigate misuse, the benchmark excludes subjective or toxic content, and all assets will be released under an open-access license with comprehensive documentation. BrowseComp-ZH promotes the development of multilingual and culturally robust AI systems by targeting the complex landscape of the Chinese web. It enables more rigorous evaluation of LLMs' retrieval and reasoning abilities, with downstream benefits in education, healthcare, and information verification. While the dataset itself poses low misuse risk, we recognize that more capable browsing agents may amplify disinformation or surveillance risks. We therefore encourage responsible deployment practices and transparency in downstream applications.

**Reproducibility Statement**   To facilitate reproducibility and further research, we will publicly release the complete BrowseComp-ZH benchmark, including: 1) The full set of 289 question-answer pairs along with domain annotations and associated constraints; 2) Annotation and verification guidelines used by human annotators, including templates and rejection criteria; 3) Evaluation scripts for accuracy and calibration error, including confidence binning and GPT-based grading prompts; 4) Model inference scripts and instructions for replicating our decoding and grading setup for both open-source and API-based models. For human evaluation, we provide anonymized results and full task instructions, ensuring transparency in baseline measurement. All released assets will be documented and hosted in a public repository, enabling both replication of our results and extension of the benchmark to new systems.

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

## A CONCERN ABOUT SMALL DATASET SIZE

We acknowledge that BrowseComp-ZH comprises only 289 examples, a scale smaller than the original English-language BrowseComp. However, this compact size stems not from oversight, but from intentional design choices rooted in the complexity of constructing a high-fidelity Chinese benchmark. To ensure that each question is both non-trivial and uniquely answerable, we adopt a labor-intensive reverse-design pipeline, with each finalized question requiring over 30 minutes on average for authoring, validation, and cross-agent auditing. Under these constraints, scaling to thousands of examples—as done in English benchmarks—is prohibitively expensive for academic settings. **Crucially, dataset scale is not the sole determinant of benchmarking value. Several widely adopted challenge-style benchmarks follow a similar philosophy of emphasizing depth and discriminative power over raw size.** These include:

- **HumanEval** Li & Murr (2024) (164 programming tasks),
- **WSC-273** Levesque et al. (2012) (273 pronoun resolution examples),
- **MATH-500** Hendrycks et al. (2021) (500 math problems),
- **SuperGLUE-CB** Wang et al. (2019) (250 examples per split),
- **MMSearch** Jiang et al. (2024) (300 manually curated web queries).

BrowseComp-ZH follows this tradition, offering high diagnostic utility despite its compact size.

Moreover, we introduce two key design enhancements over the original BrowseComp:

1. **Cross-platform retrievability filtering:** Each question is crafted to be unretrievable from the first page of Baidu, Bing, and Google, ensuring platform-independent difficulty.

2. **Multi-agent uniqueness verification:** Top-tier models such as DeepResearch, O1, and Gemini are prompted to answer each question. Their outputs are manually vetted to ensure that only questions with a single, verifiable correct answer are retained.

Despite its modest scale, BrowseComp-ZH induces wide and reliable performance gaps across systems. For instance, accuracy ranges:

- From 22.5% to 6.2% among open-source LLMs (a 16-point spread),
- From 28.9% to 7.0% among closed-source LLMs (22 points),
- From 42.9% to 4.5% across AI search agents (38+ points).

Standard deviation across three seeds is consistently under 1.6%, underscoring the benchmark's robustness and reproducibility. These results confirm that BrowseComp-ZH reliably distinguishes models based on their browsing and reasoning capabilities in a non-English, real-world setting. Rather than replicating the scale of prior benchmarks, it fills a critical gap: offering a focused, high-precision stress test for evaluating multilingual agentic systems under the linguistic, cultural, and infrastructural complexities of the Chinese web.

## B    USE OF LLMS

In this work, LLMs were used in two distinct capacities:

1. **As evaluation subjects.** The core contribution of this study is a comprehensive assessment of LLMs and AI search agents on the proposed BrowseComp-ZH benchmark. We systematically evaluate more than 20 systems—including open-source models, closed-source APIs, and retrieval-augmented agents—to draw empirical conclusions about their web browsing and reasoning capabilities in the Chinese web context.

2. **As writing assistants.** During the preparation of this manuscript, we used ChatGPT to assist with language polishing and grammar correction. All scientific content, experimental design, and analysis were authored and validated by the human researchers.

## C    INSTRUCTIONS FOR MODEL PREDICTION

In the evaluation, we followed the BrowseComp instructions for model predictions. Additionally, for both open-source and closed-source models, which may exhibit a refusal to answer (often stating the lack of search capabilities), we included an instruction that prompts the models to rely on their intrinsic knowledge for providing answers.

## D    INSTRUCTION FOR GRADING

We adopt the same grading prompt as used in (Wei et al., 2025) and employ GPT-4o for the grading process.

## E    QUALITATIVE ANALYSIS

In this section, we present a qualitative analysis of the performance of four leading AI models, selected from over 20 evaluated systems, including OpenAI DeepResearch, Perplexity, Doubao (Deep Search), and OpenAI. We focus on their performance when handling queries from the BrowseComp-ZH dataset. Our analysis highlights specific error patterns across various topics, which illustrate limitations of these AI systems.

```
Instruction for Open-source/Closed-source Model:

ZH:
如果你回复的问题需要借助外部资源，请根据你自身的知识储备给出具体答案，而不是拒答后让用户自行查询。
你的回复应遵循以下格式：
Explanation: {你对最终答案的解释}
Exact Answer: {你简洁的最终答案}
Confidence: {你对答案的置信度得分在 0% 到 100% 之间}

问题：
{问题}

EN:
If the question you are responding to may require external resources, please provide a concrete answer based on your own
knowledge base, rather than refusing to answer and directing the user to search independently.
Your response should follow the format below:
Explanation: {Your reasoning behind the final answer}
Exact Answer: {A concise statement of your final answer}
Confidence: {Your confidence in the answer, on a scale from 0% to 100%}

Question:
{Question}
```

```
Instruction for AI Search Product:

ZH:
你的回复应遵循以下格式：
Explanation: {你对最终答案的解释}
Exact Answer: {你简洁的最终答案}
Confidence: {你对答案的置信度得分在 0% 到 100% 之间}

问题：
{问题}

EN:
Your response should follow the format below:
Explanation: {Your reasoning behind the final answer}
Exact Answer: {A concise statement of your final answer}
Confidence: {Your confidence in the answer, on a scale from 0% to 100%}

Question:
{Question}
```

Figure 4: Instruction for model prediction.

For each model, we have compiled a set of failure cases, which are presented in figures ranging from 6 to 15. These figures showcase the models' responses to specific queries related to topics such as ancient Chinese music, traditional Chinese art forms, and video games. For each error case, both the original Chinese and the translated English versions are provided. The inclusion of the English version is intended to help non-native Chinese readers understand the error cases, rather than using the English version to test the model. Furthermore, the incorrect parts of the model's output are marked in red text, and the yellow highlighted portions in the figures indicate the basis on which the model's output was judged to be incorrect.

Figures 8 and 9 showcase failure cases of OpenAI DeepResearch when handling queries related to video games. These errors include a lack of understanding of game mechanics, characters, or plot points. Additionally, Figure 6 and Figure 7 highlight similar failures in responses from other AI systems. These models often provide inaccurate or incomplete information, suggesting a limited ability to engage with the rich and complex universe of modern video games.

Figures 10 and 11 focus on the models' handling of queries about music. These queries, which demand knowledge of historical contexts and cultural specifics, reveal several common errors: misidentification of song titles, incorrect associations, and misunderstandings of musical terminology. In these cases, the models often fail to account for the subtle nuances of music, leading to inaccurate or misleading responses.

Judge whether the following [response] to [question] is correct or not based on the precise and unambiguous [correct_answer] below.

[question]: {question}
[response]: {response}

Your judgement must be in the format and criteria specified below:

extracted_final_answer: The final exact answer extracted from the [response]. Put the extracted answer as 'None' if there is no exact, final answer to extract from the response.
[correct_answer]: {correct_answer}

reasoning: Explain why the extracted_final_answer is correct or incorrect based on [correct_answer], focusing only on if there are meaningful differences between [correct_answer] and the extracted_final_answer. Do not comment on any background to the problem, do not attempt to solve the problem, do not argue for any answer different than [correct_answer], focus only on whether the answers match.

correct: Answer 'yes' if extracted_final_answer matches the [correct_answer] given above, or is within a small margin of error for numerical problems. Answer 'no' otherwise, i.e. if there if there is any inconsistency, ambiguity, non-equivalency, or if the extracted answer is incorrect.

confidence: The extracted confidence score between 0% and 100% from [response]. Put 100 if there is no confidence score available.

Figure 5: Prompt for model grading.

Finally, Figures 12 and 13 illustrate the challenges faced by OpenAI DeepResearch when responding to queries about traditional Chinese art forms. These failure cases demonstrate misunderstandings of historical context and errors in identifying key concepts, such as the misclassification of specific art styles or periods. Similarly, figures 14 and 15 present failure cases of other AI systems in the same domain. These models also struggle with the accurate identification of traditional Chinese art forms, often confusing different artistic styles or misattributing them to incorrect historical periods. The highlighted errors provide a clearer picture of where the models' knowledge gaps lie in terms of Chinese cultural heritage and history.

---

主题：电子游戏

---

问题：一款恋爱冒险游戏的男主有一个真名一个假名，该游戏获得过游戏奖项，其男主配音演员小时候学过古典芭蕾，请问这个游戏的名称是？

答案：《近月少女的礼仪》

---

**Perplexity**：Explanation: 在提供的搜索结果中，没有完全符合所有要求的游戏信息。虽然《恋想关系》和《Wizards Complex》都是获奖的恋爱冒险游戏，但搜索结果没有明确表明这些游戏的男主角有真假名字，也没有证据表明任何游戏的男主角配音演员学过古典芭蕾。搜索结果中唯一提到学习芭蕾的是女性声优竹内顺子，不符合问题要求的男主配音条件。

Exact Answer: 根据提供的搜索结果无法确定符合全部条件的游戏名称。

Confidence: 20%                                                                  错误原因：无输出答案

---

**Doubao (DeepSearch)**：Explanation: 该游戏的男主角冈部伦太郎拥有真名与化名 "凤凰院凶真"，其配音演员宫野真守在童年时期学习过古典芭蕾。《命运石之门》作为经典恋爱冒险游戏，不仅在 2011 年获得 Fami 通 39 分进入白金殿堂，还凭借独特的时间旅行题材和深度剧情成为现象级作品。宫野真守在采访中曾提及舞蹈训练经历，结合其角色塑造的细腻表现，进一步印证了这一关联。

Exact Answer: 命运石之门

Confidence: 95%                                                                  错误原因：配音演员没有学过古典芭蕾

---

**OpenAI O1**：Explanation:
根据题目所给信息，"一款恋爱冒险游戏（视觉小说）男主有真名和假名，并且该游戏曾荣获游戏奖项，男主的配音演员小时候学过古典芭蕾"，符合这些特征的作品是《Steins;Gate》。
• 男主角冈部伦太郎在游戏中自称 "凤凰院凶真" (Hououin Kyouma)，即真名与假名并存。
• 《Steins;Gate》作为 5pb. 与 Nitro+ 联合制作的 "科学 ADV" 系列作品，曾获得包括 Fami 通等在内的多项游戏大奖。
• 冈部伦太郎的声优宫野真守从小就在剧团接受表演训练，他本人确实学习过古典芭蕾。

Exact Answer: 《Steins;Gate》

Confidence: 90%                                                                  错误原因：配音演员没有学过古典芭蕾

---

Figure 6: Failure cases (Chinese) of the other three AI systems in responding to a query about a video game.

---

**Topic: Video Game**

---

Question: The male protagonist of a romance adventure game has both a real name and a fake name. The game has won gaming awards, and the voice actor of the male protagonist learned classical ballet when he was a child. What is the name of this game?

Answer: The Etiquette of the Near-Moon Girl

---

**Perplexity**: Explanation: In the provided search results, there is no game information that fully meets all the requirements. While Ren'ai Kankei and Wizards Complex are both award-winning romance adventure games, the search results do not explicitly indicate that the male protagonists of these games have real and fake names, nor is there any evidence that the voice actors for these male protagonists studied classical ballet. The only mention of ballet training in the search results refers to the female voice actor, Junko Takeuchi, which does not meet the required condition for the male protagonist's voice actor.

Exact Answer: Based on the provided search results, the name of a game that meets all the conditions cannot be determined.

Confidence: 20%                                   **Error Analysis:** No answer found.

---

**Doubao (DeepSearch)** : Explanation: The male protagonist of this game, Okabe Rintarō, has both a real name and the alias "Hōōin Kyōma." His voice actor, Mamoru Miyano, studied classical ballet during his childhood. Steins;Gate, as a classic romance adventure game, not only earned a 39/40 score from Famitsu in 2011, entering the Platinum Hall of Fame, but also became a phenomenon with its unique time-travel theme and deep storyline. In interviews, Mamoru Miyano has mentioned his dance training experience, and this, combined with his delicate portrayal of the character, further supports this connection.

Exact Answer: Steins;Gate

Confidence: 95%            **Error Analysis:** The voice actor did not study classical ballet.

---

**OpenAI O1**: Explanation: Based on the information provided in the question, "a romance adventure game (visual novel) where the male protagonist has both a real name and an alias, the game has won awards, and the male protagonist's voice actor studied classical ballet in childhood," the work that matches these characteristics is Steins;Gate.

The male protagonist, Okabe Rintarō, in the game calls himself "Hōōin Kyōma," thus having both a real name and an alias.

Steins;Gate, as a "Science ADV" series work co-produced by 5pb. and Nitro+, has won numerous game awards, including the Famitsu award.

Okabe Rintarō's voice actor, Mamoru Miyano, received performance training from a young age in a theater troupe and indeed studied classical ballet.

Exact Answer: Steins;Gate

Confidence: 90%            **Error Analysis:** The voice actor did not study classical ballet.

---

Figure 7: Failure cases (translated) of the other three AI systems in responding to a query about a video game.

972
973
974
975
976
977
978
979
980
981
982
983
984
985
986
987
988
989
990
991
992
993
994
995
996
997
998
999
1000
1001
1002
1003
1004
1005
1006
1007
1008
1009
1010
1011
1012
1013
1014
1015
1016
1017
1018

---

主题：电子游戏

问题：一款恋爱冒险游戏的男主有一个真名一个假名，该游戏获得过游戏奖项，其男主配音演员小时候学过古典芭蕾，请问这个游戏的名称是？

答案：《近月少女的礼仪》

**OpenAI DeepResearch**：命运石之门（Steins;Gate）

基本信息

开发与发行：由日本5pb.和Nitro+联合开发的科学冒险类恋爱视觉小说游戏。最初于2009年10月15日在Xbox 360平台发行，随后移植到Windows、PSP、PS3、PS Vita、iOS等多个平台。

游戏类型：融合了科幻悬疑与恋爱要素的文字冒险游戏，拥有多重分支剧情和结局。

故事背景：讲述一群秋叶原年轻科研宅意外发明时间短信技术，引发时间线变化的故事。其中既有紧张的科幻悬疑，也穿插着角色之间细腻的情感线。

男主角的姓名（真名与假名）

真名：冈部伦太郎（Okabe Rintarō），绰号"冈伦"。他是东京电机大学一年级学生，也是自封的"未来道具研究所"实验小组的创立者。

假名/代号：自称**"凤凰院凶真"**（Hōōin Kyōma）作为自己的中二病式假名。冈部经常以"凤凰院凶真"的身份夸张言行，伪装成疯狂科学家。这一假名实际上是他给自己营造的第二身份，与真实姓名形成对比。一些角色（如女主牧濑红莉栖）最初认为这是他在开玩笑，但"凤凰院凶真"的称呼贯穿剧情，体现了冈部中二且逃避现实的一面。

男主角配音演员及童年经历

配音演员（日语版）：宫野真守。冈部伦太郎在游戏及其动画改编中均由声优宫野真守出演。他是日本知名男性声优、演员和歌手，代表作包括《死亡笔记》的夜神月等。

童年经历：宫野真守自幼便参与表演训练，7岁时即加入剧团向日葵成为童星开始演艺活动。在剧团的培养下，他从小接受了全面的表演教育，其中包括声乐和舞蹈训练。据其访谈和业界资料透露，宫野小时候曾学习古典芭蕾等舞蹈，为日后在舞台剧和声优事业上的肢体表现打下了基础。这一童年经历赋予了他优异的节奏感和肢体控制力，使他在配音时能更好地拿捏角色的情感张力。宫野真守青春期还活跃于音乐剧舞台（如网舞《网球王子》音乐剧），足见其舞蹈功底和表演才能出众。

游戏获得的奖项信息

业内奖项：*《命运石之门》*在发售后广受好评，获得多项业界荣誉。例如，它荣获了2009年度日本Fami通游戏大奖的优秀赏（相当于年度优秀游戏奖）。这一奖项肯定了该作在剧情深度和创新玩法上的出色表现。

石之门》*符合题目所有条件：男主角冈部伦太郎有真名和"凤凰院凶真"假名；其日语声优宫野真守从小接受古典芭蕾等表演训练；游戏本身亦荣获业界大奖，备受赞誉。这一作品无论在剧情、角色还是制作成就上都堪称佳作，非常值得一试。

参考来源：

【命运石之门 游戏介绍】维基百科条目：命运石之门

【男主角人名与假名】萌娘百科：冈部伦太郎角色资料

【配音演员及经历】维基百科：宫野真守人物条目（童星出道经历）

【游戏奖项】维基百科：命运石之门（评价部分，Fami通优秀赏等）

错误原因：配音演员没有学过古典芭蕾

---

Figure 8: Failure case (Chinese) of OpenAI DeepResearch in responding to a query about a video game.

1019
1020
1021
1022
1023
1024
1025

---

**Topic: Video Game**

Question: The male protagonist of a romance adventure game has both a real name and a fake name. The game has won gaming awards, and the voice actor of the male protagonist learned classical ballet when he was a child. What is the name of this game?

Answer: The Etiquette of the Near-Moon Girl

---

**OpenAI DeepResearch**: Steins;Gate

Basic Information. Development and Release: Steins;Gate is a scientific adventure romance visual novel game co-developed by 5pb. and Nitro+. It was originally released on October 15, 2009, for the Xbox 360 platform, and later ported to multiple platforms, including Windows, PSP, PS3, PS Vita, iOS, and others. Game Type: A text-based adventure game that blends sci-fi suspense and romance elements, featuring multiple branching storylines and endings.

Story Background: The story follows a group of young researchers in Akihabara who accidentally invent a technology to send text messages through time, triggering changes in the timeline. The narrative features both tense sci-fi suspense and nuanced emotional lines between the characters.

Male Protagonist's Name (Real Name and Alias)

Real Name: Okabe Rintarō (冈部伦太郎), nickname "Okarin." He is a first-year student at Tokyo University of Electro-Communications and the self-proclaimed founder of the "Future Gadget Laboratory." Alias/Codename: He uses the middle-schooler-like alias "Hōōin Kyōma" (凤凰院凶真), adopting exaggerated behavior and pretending to be a mad scientist. This alias serves as his second identity, contrasting with his real name. Some characters (such as the heroine, Makise Kurisu) initially think he is joking, but the alias "Hōōin Kyōma" runs throughout the story, reflecting Okabe's delusional and escapist side.

Voice Actor and Childhood Experience. Voice Actor (Japanese version): Mamoru Miyano. Okabe Rintarō is voiced by Mamoru Miyano in both the game and its animated adaptations. Miyano is a well-known Japanese male voice actor, actor, and singer, with notable roles including Light Yagami in Death Note.

Childhood Experience: Miyano started performing at a young age, joining the Sunflower Theater troupe at the age of 7 and beginning his career as a child star. Under the troupe's guidance, he received comprehensive training in performance, including vocal and dance education. According to interviews and industry information, Miyano studied classical ballet during his childhood, laying the foundation for his future physical performance skills in stage plays and voice acting. This experience provided him with excellent rhythm and body control, allowing him to better convey emotional tension in his voice acting. During his teenage years, Miyano also performed in musical theater (such as in the Prince of Tennis musical), showcasing his exceptional dance and acting abilities.

Awards the Game Has Received. Industry Awards: Steins;Gate received widespread acclaim after its release, winning several industry honors. For example, it won the "Excellence Award" at the 2009 Famitsu Game Awards (equivalent to the Game of the Year Award), which recognized the game for its deep narrative and innovative gameplay. Steins;Gate meets all the criteria in the question: the male protagonist, Okabe Rintarō, has both a real name and the alias "Hōōin Kyōma"; the Japanese voice actor, Mamoru Miyano, received training in classical ballet and other performance arts from a young age; and the game itself won prestigious industry awards and received widespread praise. This work is a masterpiece in terms of its story, characters, and production, making it highly recommended for anyone to try.

Reference Sources:

Steins;Gate Game Introduction: Wikipedia entry on Steins;Gate

Male Protagonist's Name and Alias: Moe Daughter Encyclopedia: Okabe Rintarō character information

Voice Actor and Experience: Wikipedia entry on Mamoru Miyano (child star debut)

Game Awards: Wikipedia entry on Steins;Gate (evaluation section, Famitsu Excellence Award, etc.)

**Error Analysis:** The voice actor did not study classical ballet.

---

Figure 9: Failure case (translated) of OpenAI DeepResearch in responding to a query about a video game.

主题：音乐

问题：一个专辑名称与中国古代一本古籍中记载过的一次流星雨辐射点位置对应星座名称相同，该专辑中的一首音乐与一部游戏中的某个角色技能有关，请问这首音乐名称是？

答案：《原色》

**OpenAI DeepResearch**：专辑名称：《天琴座+》（Lyra+）。这个专辑名称取自"天琴座"星座名称，正是中国古籍《左传》记载的一次流星雨（公元前687年四月天琴座流星雨）的辐射点所在星座。换言之，"天琴座"对应的是英文 Lyra 星座，古籍中那场流星雨就源自该星座方向。(......) 专辑收录曲目之一**《闪亮》**（"Shining"）就是耀嘉音的代表作，在游戏宣传片及角色演出中作为背景曲出现。这一曲目与耀嘉音的技能演出紧密结合，体现角色以歌声（闪耀的星光）战斗的设定。

最终曲名：《闪亮》（Shining）。该曲既是专辑《天琴座+》中的一首歌曲，也是游戏中耀嘉音角色演唱并用于技能展示的曲目，二者相互关联。

参考资料：《左传》流星雨记载；绝区零《天琴座+》专辑信息；耀嘉音角色技能机制；耀嘉音歌曲《闪亮》发布信息。 错误原因：《闪亮》与技能无关

**Perplexity**: Explanation: 通过分析搜索结果，我发现中国古籍《春秋》记载了一次天琴座流星雨；《绝区零》游戏中有一个名为"天琴座+"的专辑；专辑中的"闪亮"这首歌与游戏角色耀嘉音的闪避技能《小华尔兹》在功能上相关。

Exact Answer: 闪亮

Confidence: 85% 错误原因：《闪亮》与技能无关

**Doubao (DeepSearch)**: Explanation: 中国古代《左传》记载的鲁庄公七年（公元前687年）四月流星雨，被确认为天琴座流星雨的最早记录35。专辑《天琴座+》由 miHoYo 旗下音乐团队 HOYO-MiX 制作，是游戏《绝区零》的官方音乐专辑，专辑名称与天琴座直接对应715。该专辑收录的曲目《乐园游梦记》与《绝区零》中代理人耀嘉音的角色设定相关，而耀嘉音的技能 "乐园游梦记" 在游戏中表现为通过音乐操控幻境的能力，与曲目名称及风格高度契合1517。

Exact Answer: 乐园游梦记

Confidence: 85% 错误原因：与技能无关

**OpenAI O1**: Explanation: 根据题目线索："中国古代一本古籍中记载过的一次流星雨的辐射点所在星座"较为有名的便是"摩羯座""狮子座""天琴座"等。其中，周杰伦于 2008 年发行过一张名为《魔杰座》的专辑，其英文名正是 "Capricorn"，对应的便是"摩羯座 (Capricornus)"。而摩羯座的流星雨（α-摩羯座流星雨）在中国古代典籍中也有过记载。接下来，再看该专辑收录曲目，是否有与某款游戏中角色技能相关联的歌名。周杰伦《魔杰座》专辑中的曲目包括《龙战骑士》《给我一首歌的时间》《蛇舞》《花海》《魔术先生》《说好的幸福呢》《兰亭序》《流浪诗人》《时光机》《乔克叔叔》《稻香》。其中最常被拿来与游戏技能关联的是专辑首曲《龙战骑士》。比如在部分游戏（含 MOBA、RPG 等）中，"Dragon Knight""Dragon Rider"之类的角色/技能命名相当常见；亦有玩家会将《龙战骑士》与某些游戏里"化身巨龙"或"召唤巨龙"等大招技能联系起来。因此，题目所问"该专辑中与某个游戏角色技能相关的那首音乐"最常被认定就是《龙战骑士》。

Exact Answer: 《龙战骑士》

Confidence: 85% 错误原因：专辑名不对

Figure 10: Failure cases (Chinese) of four models in responding to a query about ancient Chinese music, highlighting inaccuracies in song title identification, irrelevant associations, and context misinterpretation.

---

**Topic: Music**

Question: The name of an album corresponds to the name of the constellation that was recorded as the radiant point of a meteor shower in an ancient Chinese text. One of the tracks in this album is related to a character's skill in a game. What is the name of this track?

Answer: "Original Color" (《原色》)

---

**OpenAI DeepResearch**: Album Name: "Lyra+". The name of this album comes from the "Lyra" constellation, which is the radiant point of a meteor shower recorded in the ancient Chinese text Zuo Zhuan (The Zuo Tradition) in April 687 BC. In other words, "Lyra" corresponds to the constellation Lyra in English, and the meteor shower in the ancient text originated from the direction of this constellation. One of the tracks on the album, "Shining", is a representative work of the artist Yao Jiayin. It appears as a background music in the game's promotional videos and character performances. This track is closely related to Yao Jiayin's character skill performance, which reflects the setting where the character fights using her voice (shining starlight). Final Track Name: "Shining" This track is not only a song on the album Lyra+ but also the song performed by the character Yao Jiayin in the game, used for skill demonstrations. The two are interconnected.

Reference Materials: Zuo Zhuan meteor shower records; Absolute Zero game's Lyra+ album information; Yao Jiayin's character skill mechanics; release information for Yao Jiayin's song Shining.

**Error Analysis:** "Shining" is unrelated to any character's skill.

---

**Perplexity**: Explanation: By analyzing the search results, I found that the ancient Chinese text *Spring and Autumn Annals* recorded a meteor shower from the Lyra constellation. In the game *Absolute Zero*, there is an album called "Lyra+" and the song "Shining" in the album is functionally related to the dodge skill "Little Waltz" of the game character Yao Jiayin.

Exact Answer: Shining

Confidence: 85%       **Error Analysis:** "Shining" is unrelated to any character's skill.

---

**Doubao (DeepSearch)** : Explanation: The meteor shower recorded in the 7th year of Duke Zhuang of Lu (687 BC) in April in the ancient Chinese text Zuo Zhuan is confirmed to be the earliest recorded Lyra meteor shower. The album Lyra+ was produced by HOYO-MiX, the music team under miHoYo, and is the official soundtrack for the game Absolute Zero. The album name directly corresponds to the Lyra constellation. One of the tracks on the album, Dreams of the Paradise, is related to the character setting of the agent Yao Jiayin in Absolute Zero, and Yao Jiayin's skill "Dreams of the Paradise" in the game manifests as the ability to manipulate illusions through music, which is highly in tune with the track's title and style.

Exact Answer: Dreams of the Paradise       **Error Analysis:** "Dreams of the Paradise" is unrelated to any character's skill.

Confidence: 85%

---

**OpenAI O1** : Explanation: Based on the clues in the question, "the constellation where the radiant point of a meteor shower recorded in an ancient Chinese text is located," some well-known constellations include "Capricorn," "Leo," and "Lyra." Among them, Jay Chou released an album in 2008 called Capricorn, which corresponds to the "Capricornus" constellation. The meteor shower associated with Capricorn (α-Capricorni meteor shower) was also recorded in ancient Chinese texts. Next, we examine the tracklist of this album to see if any song titles are related to character skills in a certain game. The tracks in Jay Chou's Capricorn album include "Dragon Knight," "Give Me a Song's Time," "Snake Dance," "Sea of Flowers," "Mr. Magic," "Where's the Happiness We Promised," "Lanting Preface," "Wandering Poet," "Time Machine," "Uncle Joker," and "Rice Aroma." The track most commonly associated with game skills is the album's opening song, "Dragon Knight." For example, in some games (including MOBA, RPG, etc.), character/skill names like "Dragon Knight" or "Dragon Rider" are quite common; players often associate "Dragon Knight" with ultimate skills like "Transform into a Dragon" or "Summon a Dragon." Therefore, the song most often recognized as being related to a game character's skill in this album is Dragon Knight.

Exact Answer: Dragon Knight

Confidence: 85%       **Error Analysis:** The album name is incorrect.

---

Figure 11: Failure cases (translated) of four models in responding to a query about ancient Chinese music, highlighting inaccuracies in song title identification, irrelevant associations, and context misinterpretation.

---

主题：艺术

问题：在中国传统艺术中，有一种特殊的绘画形式，起源于元代，盛行于清末，传说是由一位古代知名画家在酒后兴起所创。这种艺术形式在2010~2015年之间被列入某省级非物质文化遗产名录。绘制这种艺术形式需要画家精通各种画法，并且要善书各种字体。请问这种艺术形式是什么？

答案：锦灰堆

---

**OpenAI DeepResearch**：指画（手指画）传统艺术形式

起源与发展：指画，又称手指画、指头画或指墨画，是以手指代替毛笔在纸绢上蘸墨作画的一种中国传统美术形式。相传唐代画家张璪曾在酒后不用笔而以手蘸墨作画，被后世视为指画技法的鼻祖；他曾说"指头画，画之极致"，强调以手代笔作画的独特魅力。指画这一技法在元代已有所发展，到了清代晚期达到鼎盛** **。清初以来出现了专攻指画的名家，如清代指画大师高其佩等，使这种画法流派逐渐成熟。

艺术特点：指画的技法独特，需要画家以手指、手掌乃至指甲来代替毛笔运笔作画。由于直接以手运墨，创作时对力度和墨色的控制要求极高，画家往往要具备扎实的中国画功底，精通工笔与写意等多种画法。同时，指画作品常辅以题款诗词，要求创作者擅长多种书法字体，以手指运笔写出篆、隶、行、草等书法效果，从而达到画中有字、字中有画的艺术境界。因此，一位优秀的指画画家往往既是画家也是书法家。清代高其佩便以指画闻名于世，他幼年先学传统工笔画，后在梦境启发下改用手指作画达数十年之久，不再用笔** **。在实践中，高其佩融合了山水、人物、花鸟等画科的笔法，并将书法用笔之道融入指画，使指画艺术臻于化境。

非遗保护现状

作为中国民间艺术的瑰宝，指画艺术具有独特的人文和艺术价值。近年来，各地纷纷将指画列入非物质文化遗产名录加以保护。早在2006年，辽宁省便将"指画艺术"确定为首批省级非物质文化遗产项目。此后，包括湖北省在内的多个省市也相继将指画列入非遗名录。比如，湖北省于2009年公布的第二批省级非遗名录中，就将"手指画"列为传统美术项目进行保护** **。通过非遗认定和传承人培养等措施，这一源远流长的传统画艺得以在现代焕发新的生机。

参考资料：

【21】张恺新. 辽宁非遗 | 指画艺术：以指代笔 风趣横生. 学习强国平台, 2019年9月6日.

【26】中国日报网江西频道. 肖洪波指掌画列入宜春非物质文化遗产名录, 2021年11月15日.

【42】Wikipedia. Gao Qipei. (高其佩指画传统及张璪指画先例)

【45】【37】湖北省文化和旅游厅. 湖北省非物质文化遗产代表性项目名录 (第二批), 2009年公布.

错误原因：时间不对，题目说2010-2015，答案给的2009和2006

---

Figure 12: Failure case (Chinese) of OpenAI DeepResearch in responding to a query about traditional Chinese art forms, showing misunderstandings of historical context and incorrect identification of key concepts.

---

**Topic: Art**

Question: In traditional Chinese art, there is a unique form of painting that originated in the Yuan Dynasty and became popular during the late Qing Dynasty. It is said to have been created by a famous ancient painter who was inspired by alcohol. Between 2010 and 2015, this art form was included in the provincial intangible cultural heritage list of a certain region. To paint in this style, artists must be proficient in various painting techniques and skilled in writing different types of calligraphy. What is this art form called?

Answer: Heaps of Brocade and Ash

---

**OpenAI DeepResearch**: Finger Painting (Traditional Art Form)

Origin and Development: Finger painting, also known as hand painting, finger-tip painting, or ink finger painting, is a traditional Chinese art form where the artist uses their fingers instead of a brush to apply ink on paper or silk. According to legend, the Tang Dynasty artist Zhang Zao once painted with his fingers after drinking wine, without using a brush, and this is considered the origin of the finger painting technique. He is said to have remarked, "Finger painting, the utmost in painting," emphasizing the unique charm of using the hand instead of a brush for painting. This technique began to develop during the Yuan Dynasty and reached its peak during the late Qing Dynasty. Since the early Qing period, renowned masters specializing in finger painting, such as the Qing Dynasty finger painting master Gao Qipei, contributed to the gradual maturation of this painting style.

Artistic Characteristics: The technique of finger painting is unique and requires the artist to use their fingers, palms, and even nails to replace a brush in applying ink. Since the artist directly applies the ink with their hand, the control of force and ink color is crucial. Therefore, the artist usually needs a solid foundation in traditional Chinese painting, mastering both fine brushwork and freehand techniques. Additionally, finger painting works are often accompanied by inscriptions or poetry, requiring the artist to be skilled in various calligraphy styles, writing seal script, clerical script, regular script, and cursive script with their fingers, creating an artistic realm where both words and paintings exist together. As a result, an excellent finger painting artist is often both a painter and a calligrapher. Gao Qipei, a well-known Qing Dynasty artist, became famous for his finger painting. He first studied traditional meticulous painting in his childhood, then switched to finger painting after being inspired by dreams, a practice he continued for decades, never using a brush again. In practice, Gao Qipei integrated techniques from landscapes, figures, and flower-and-bird paintings, blending calligraphy principles into his finger painting, perfecting the art form.

Status of Intangible Cultural Heritage Protection: As a treasure of Chinese folk art, finger painting holds unique cultural and artistic value. In recent years, many regions have included finger painting in their lists of intangible cultural heritage for protection. As early as 2006, Liaoning Province identified "Finger Painting Art" as one of the first batch of provincial intangible cultural heritage projects. Subsequently, many provinces and cities, including Hubei Province, added finger painting to their intangible heritage lists. For example, in the second batch of provincial intangible cultural heritage lists announced in 2009, Hubei Province included "Finger Painting" as part of its traditional art protection efforts. Through measures like the recognition of intangible heritage and the training of inheritors, this long-standing traditional art has been revitalized in modern times.

References:

【21】Zhang Kaixin. Liaoning Intangible Heritage | Finger Painting Art: Replacing Brushes with Fingers, Fun and Engaging. Xuexi Qiangguo Platform, September 6, 2019.

【26】China Daily Jiangxi Channel. Xiao Hongbo's Finger Palm Painting Included in Yichun's Intangible Cultural Heritage List, November 15, 2021.

【42】【37】Hubei Provincial Department of Culture and Tourism. Hubei Province Intangible Cultural Heritage Representative Project List (Second Batch), Announced in 2009.

**Error Analysis:** The time is incorrect; the question specifies 2010-2015, but the answer provided refers to 2009 and 2006.

---

Figure 13: Failure case (translated) of OpenAI DeepResearch in responding to a query about traditional Chinese art forms, showing misunderstandings of historical context and incorrect identification of key concepts.

主题：艺术

问题：在中国传统艺术中，有一种特殊的绘画形式，起源于元代，盛行于清末，传说是由一位古代知名画家在酒后兴起所创。这种艺术形式在2010~2015年之间被列入某省级非物质文化遗产名录。绘制这种艺术形式需要画家精通各种画法，并且要善书各种字体。请问这种艺术形式是什么？

答案：锦灰堆

**Perplexity**：Explanation: 通过分析搜索结果，我找到了关于元代画家酒后作画的丰富信息，特别是元四家中的三位大师。但没有发现一种完全符合题目所有条件的艺术形式。最接近的是文人画或泼墨山水，它们确实需要画家精通各种画法和字体，但无法确定它们是否在2010-2015年间被列入省级非物质文化遗产名录。

Exact Answer: 文人画或泼墨山水

Confidence: 60%          错误原因：无法确定时间

**Doubao (DeepSearch)**：Explanation: 这个传统的中国艺术形式是指画（指画／指墨画）。它起源于清代（而不是元代，可能是一个常见的误解），由高其佩这位著名的画家开创，他以其酒后即兴创作而闻名。指画在清朝晚期获得了广泛的传播，并于2014年被四川省列为省级非物质文化遗产，名为"峨眉山指画"。这种技法要求画家精通多种笔法和书法，因为艺术家使用手指、指甲和手掌来创造独特的质感和形式。虽然元代的起源可能是错误的归因，但其他的标准与指画完全吻合。

Exact Answer: 指画

Confidence: 90%          错误原因：指画不起源于元代

**OpenAI O1**：Explanation:
据传最早的指画（亦称"指墨画"）可追溯至元代，到了清代时，由高其佩等人的大力推广而兴盛。指画的民间传说称，此画法起源于某位古代名家酒后偶然失落画笔，情急之下便以手指蘸墨作画，从而形成独特的指画技法。指画的创作要求画家既要熟悉多种绘画技法，也要擅长不同书体的书法，以手指、手掌乃至指甲来表现笔意。2010~2015年间，某些地区将指画列入省级非物质文化遗产名录，进一步推动了这一艺术形式的保护与传承。

Exact Answer: 指画（指墨画）

Confidence: 90%       错误原因：指画起源自唐代而不是元代

Figure 14: Failure cases (Chinese) of other AI systems in responding to a query about traditional Chinese art forms, illustrating issues such as incorrect identification of the art form and confusion about its historical origins and periods.

1350
1351
1352
1353
1354
1355
1356
1357
1358
1359
1360
1361
1362
1363
1364
1365
1366
1367
1368
1369
1370
1371
1372
1373
1374
1375
1376
1377
1378
1379
1380
1381
1382
1383
1384
1385
1386
1387
1388
1389
1390
1391
1392
1393
1394
1395
1396
1397
1398
1399
1400
1401
1402
1403

---

**Topic: Art**

---

Question: In traditional Chinese art, there is a unique form of painting that originated in the Yuan Dynasty and became popular during the late Qing Dynasty. It is said to have been created by a famous ancient painter who was inspired by alcohol. Between 2010 and 2015, this art form was included in the provincial intangible cultural heritage list of a certain region. To paint in this style, artists must be proficient in various painting techniques and skilled in writing different types of calligraphy. What is this art form called?

Answer: Heaps of Brocade and Ash

---

**Perplexity**: Explanation: By analyzing the search results, I found abundant information about Yuan dynasty painters painting after drinking, especially three masters from the Four Masters of the Yuan Dynasty. However, I did not find an art form that fully meets all the criteria of the topic. The closest are literati painting or splashed ink landscape, which indeed require the artist to be proficient in various painting techniques and calligraphy, but it is unclear whether they were included in the provincial intangible cultural heritage list between 2010 and 2015.

Exact Answer: Literati painting or splashed ink landscape

Confidence: 60%                     Error Analysis: Unable to determine the time.

---

**Doubao (DeepSearch)** :   Explanation: The traditional Chinese art form in question is finger painting (指画 / 指墨画). Originating in the Qing Dynasty (not the Yuan Dynasty as mentioned, which might be a common misunderstanding), it was pioneered by Gao Qipei, a renowned painter known for his spontaneous, alcohol-inspired creations. Finger painting gained popularity in the late Qing Dynasty and was inscribed as a provincial intangible cultural heritage in Sichuan Province in 2014 under the name "Emei Mountain Finger Painting." This technique requires mastery of diverse brushwork styles and calligraphy, as artists use fingers, nails, and palms to create unique textures and forms. While the Yuan Dynasty origin is likely a misattribution, the other criteria align perfectly with finger painting.

Exact Answer: finger painting

Confidence: 90%             Error Analysis: Finger painting did not originate in the Yuan Dynasty.

---

**OpenAI O1:** Explanation: It is said that the earliest finger painting (also known as "finger ink painting") dates back to the Yuan Dynasty, and during the Qing Dynasty, it flourished due to the vigorous promotion by figures like Gao Qipei. According to folklore, this painting method originated when a famous ancient artist, after drinking, accidentally dropped his brush. In a moment of urgency, he used his fingers dipped in ink to paint, thus creating the unique finger painting technique. The creation of finger paintings requires the artist to be familiar with various painting techniques and proficient in different styles of calligraphy, using fingers, palms, and even nails to express the brushwork. Between 2010 and 2015, certain regions included finger painting in the provincial intangible cultural heritage list, further promoting the protection and inheritance of this art form.

Exact Answer: Finger painting (Finger ink painting)     Error Analysis: Finger painting originated in the Tang Dynasty, not the Yuan Dynasty.

Confidence: 90%

---

Figure 15: Failure cases (translated) of other AI systems in responding to a query about traditional Chinese art forms, illustrating issues such as incorrect identification of the art form and confusion about its historical origins and periods.