# OpenReview forum: "BrowseComp-ZH: Benchmarking Web Browsing Ability of Large Language Models in Chinese"
_ICLR.cc/2026/Conference — Submitted to ICLR 2026_

### Official Review · Reviewer_8svw · 2025-10-30

**Soundness:** 3
**Presentation:** 2
**Contribution:** 2
**Rating:** 6
**Confidence:** 3

**Summary:**

This paper introduces BrowseComp-ZH, a benchmark for evaluating LLM agents’ web browsing and information-seeking abilities specifically within the Chinese web ecosystem. The dataset comprises 289 human-authored, multi-hop, multi-constraint questions across 11 knowledge domains, meticulously reverse-engineered from verifiable answers. The paper also provides thorough statistics, calibration analysis, multiple failure case studies, and benchmark reproducibility details.

**Strengths:**

- The paper fills a clear gap by constructing the first natively-annotated Chinese web browsing benchmark, avoiding translation artifacts and capturing authentic linguistic, cultural, and infrastructural challenges unique to the Chinese internet.
- The reverse-design pipeline is impressively meticulous, resulting in a high-quality, stress-test-style dataset.

**Weaknesses:**

- At only 289 questions, BrowseComp-ZH is considerably smaller than many contemporary benchmarks. This small scale could weaken robustness for fine-grained model assessment or pre-train/fine-tune settings in the future.
- Despite employing multi-agent and human-in-the-loop processes, the guarantee of a single unique answer still depends on current model/retrieval limitations and annotator creativity.

**Questions:**

- How were the final domain-specific question counts determined? Is there a risk of over-representation of certain topics?
- How can the annotators ensure that the difficulty of the problem is sufficient and that this benchmark has the ability to evaluate the entire RAG system?

---

> ### Author Response · Authors · 2025-11-21
> **Rebuttal to Reviewer 8svw**
>
> We thank the reviewer for the valuable feedback. We appreciate your scrutiny regarding the dataset scale and construction rigor. We address your specific concerns below:
>
> ---
>
> **Q1**: Dataset Scale and Robustness
>
> **A1**: We acknowledge that BrowseComp-ZH is smaller in scale than the original English-language BrowseComp. However, this is a result of intentional design, not oversight. The compact size is due to the complexity of creating a high-fidelity Chinese benchmark, where each question requires over 30 minutes for authoring, validation, and cross-agent auditing. Given these constraints, scaling to thousands of examples, as in English benchmarks, would be prohibitively expensive for academic settings.
>
> Importantly, the value of a benchmark is not solely determined by its size. Several well-known challenge benchmarks emphasize depth and discriminative power over raw scale, such as:
>
> - **HumanEval** (164 programming tasks)
> - **WSC-273** (273 pronoun resolution examples)
> - **MATH-500** (500 math problems)
> - **SuperGLUE-CB** (250 examples per split)
> - **MMSearch** (300 manually curated web queries)
>
> BrowseComp-ZH follows this philosophy, offering significant diagnostic utility despite its smaller size. Additionally, we introduce two key enhancements over the original BrowseComp:
>
> 1. **Cross-platform retrievability filtering:** Questions are designed to be unretrievable from the first page of Baidu, Bing, and Google, ensuring consistent difficulty across platforms.
> 2. **Multi-agent uniqueness verification:** Answers from top-tier models (DeepResearch, O1, Gemini) are manually verified to retain only questions with a single, verifiable correct answer.
>
> Despite its smaller scale, BrowseComp-ZH reveals wide performance gaps:
>
> - **Open-source LLMs**: Accuracy ranges from **22.5% to 6.2%** (16-point spread)
> - **Closed-source LLMs**: Accuracy ranges from **28.9% to 7.0%** (22-point spread)
> - **AI search agents**: Accuracy ranges from **42.9% to 4.5%** (38+ points)
>
> Standard deviation across three seeds remains under 1.6%, demonstrating the benchmark's robustness. BrowseComp-ZH offers a focused, high-precision stress test for evaluating multilingual systems in the complex context of the Chinese web, filling a critical gap in current benchmarks.
>
> ---
>
> **Q2**: Guaranteeing Answer Uniqueness
>
> **A2**: The reviewer correctly notes that perfect uniqueness is theoretically challenging. However, we employed a rigorous adversarial validation process to minimize ambiguity:
>
> * **Multi-Agent Adversarial Testing:** We did not rely solely on annotators. We deployed multiple top-tier agents (**DeepResearch, O1, Gemini**) to solve each candidate question. If these agents found any valid alternative answer that satisfied the constraints, the question was discarded. This process filtered out **115 ambiguous samples** from the initial pool.
> * **Reverse Design:** By starting with a specific fact and adding multiple constraints (temporal, spatial, categorical), the solution space is mathematically narrowed, making accidental alternative answers highly unlikely compared to forward-designed questions.
>
> ---
>
> **Q3**: Domain Selection
>
> **A3**: Regarding the selection of domains and topics:
>
> * **Expertise-Driven Selection:** To ensure questions were non-trivial, annotators selected topics based on their personal expertise and interests (e.g., a history buff writes history questions). This prevents "generic" questions that are easy to retrieve.
> * **Diverse Coverage:** While this leads to slight variations in distribution (e.g., 15.6% Film & TV vs. 3.5% Policy), the benchmark covers **11 distinct domains**, ensuring that models are tested on a wide variety of vocabulary and search intents.
>
> ---
>
> **Q4**: Ensuring Difficulty and RAG Evaluation
>
> **A4**: We employ specific mechanisms to ensure difficulty and effectively test RAG systems:
>
> * **The Triple-Engine Test:** Difficulty is empirically enforced via a strict rule: a question is retained only if the correct answer **does not appear on the first page of Baidu, Bing, or Google results**. This explicitly tests the RAG system's ability to perform multi-hop retrieval rather than simple keyword matching.
> * **Evaluating the Full RAG Pipeline:** Success on BrowseComp-ZH requires a system to (1) plan search queries, (2) retrieve fragmented evidence from the Chinese web's "walled gardens," and (3) reason over conflicting information. The extremely low accuracy of standard RAG systems (e.g., $<10\%$ for many open-source models) confirms that the benchmark effectively exposes failures in both the retrieval and reasoning components.
>
> We hope these clarifications demonstrate the rigor and utility of BrowseComp-ZH.

---

> > ### Comment · Reviewer_8svw · 2025-11-25
> > **Thanks for the response**
> >
> > I appreciate the authors for their response. Good luck!

---

### Official Review · Reviewer_yxRg · 2025-10-31

**Soundness:** 2
**Presentation:** 2
**Contribution:** 2
**Rating:** 4
**Confidence:** 5

**Summary:**

This paper introduces BrowseComp-ZH, a new, high-difficulty benchmark designed to evaluate the web browsing and information retrieval capabilities of Large Language Models (LLMs) specifically within the Chinese web ecosystem.

Key Contributions:
* Natively Constructed: It consists of 289 questions written from scratch by native Chinese speakers to reflect authentic user behavior, rather than being translated from English.

* High Difficulty: The questions are multi-hop, requiring complex reasoning and the ability to synthesize information from multiple sources.

* Rigorous Design: Questions were reverse-engineered from verified answers and passed through a two-stage quality control process to ensure they are difficult to solve and have unique answers.

**Strengths:**

1. Fills a Critical Gap: It’s the first benchmark for evaluating LLMs’ web-browsing abilities in the Chinese ecosystem, avoiding flaws of translated English benchmarks by using native Chinese questions that reflect linguistic, cultural, and Chinese web-specific traits.

2. Rigorous Construction: Its 289 multi-hop questions (11 domains) go through strict quality control—reverse-engineered from answers, tested on major search engines to ensure difficulty, and validated for answer uniqueness.

3. Insightful Evaluation: It assesses over 20 systems (open/closed-source models, commercial tools) and includes human baselines (17.6% accuracy), revealing key findings (e.g., reasoning and multi-round retrieval boost performance) to guide LLM improvement.

4. Practical Guidance: Serves as a targeted stress test for non-English web LLMs, aiding the development of multilingual AI agents for real-world Chinese web scenarios.

**Weaknesses:**

1. Small Dataset Scale: With only 289 questions, it is smaller than English counterparts like BrowseComp, though the authors note this stems from high curation costs.

2. Incomplete Answer Uniqueness: Despite strict checks, the reverse-design approach cannot fully guarantee no alternative valid answers exist for some questions. There are also some obvious bugs in the dataset that need to be fixed.

3. Compared to BrowseComp, BrowseComp-ZH is insufficiently challenging. It is essentially a simplified Chinese adaptation that lacks originality.

**Questions:**

See Weaknesses

---

> ### Author Response · Authors · 2025-11-21
> **PART I:  Dataset Scale and Answer Uniqueness**
>
> We thank the reviewer for their constructive feedback. Below, we address the concerns regarding dataset scale, uniqueness, and the benchmark's difficulty and novelty.
>
> ---
>
> **Q1**: Concerns regarding the small dataset scale.
>
> **A1**: We acknowledge that the scale of BrowseComp-ZH is smaller than the original English-language BrowseComp. However, this compact size stems not from oversight, but from intentional design choices rooted in the complexity of constructing a high-fidelity Chinese benchmark. To ensure that each question is both non-trivial and uniquely answerable, we adopt a labor-intensive reverse-design pipeline, with each finalized question requiring over 30 minutes on average for authoring, validation, and cross-agent auditing. Under these constraints, scaling to thousands of examples—as done in English benchmarks—is prohibitively expensive for academic settings.
>
> Crucially, dataset scale is not the sole determinant of benchmarking value. Several widely adopted challenge-style benchmarks follow a similar philosophy of emphasizing depth and discriminative power over raw size. These include:
>
> * **HumanEval** Li & Murr (2024) (164 programming tasks)
> * **WSC-273** Levesque et al. (2012) (273 pronoun resolution examples)
> * **MATH-500** Hendrycks et al. (2021) (500 math problems)
> * **SuperGLUE-CB** Wang et al. (2019) (250 examples per split)
> * **MMSearch** Jiang et al. (2024) (300 manually curated web queries)
>
> BrowseComp-ZH follows this tradition, offering high diagnostic utility despite its compact size. Moreover, we introduce two key design enhancements over the original BrowseComp:
>
> 1.  **Cross-platform retrievability filtering:** Each question is crafted to be unretrievable from the first page of Baidu, Bing, and Google, ensuring platform-independent difficulty.
> 2.  **Multi-agent uniqueness verification:** Top-tier models such as DeepResearch, O1, and Gemini are prompted to answer each question. Their outputs are manually vetted to ensure that only questions with a single, verifiable correct answer are retained.
>
> Despite its modest scale, BrowseComp-ZH induces wide and reliable performance gaps across systems. For instance, accuracy ranges:
>
> * From **22.5% to 6.2%** among open-source LLMs (a 16-point spread)
> * From **28.9% to 7.0%** among closed-source LLMs (22 points)
> * From **42.9% to 4.5%** across AI search agents (38+ points)
>
> Standard deviation across three seeds is consistently under 1.6%, underscoring the benchmark’s robustness and reproducibility. These results confirm that BrowseComp-ZH reliably distinguishes models based on their browsing and reasoning capabilities in a non-English, real-world setting. Rather than replicating the scale of prior benchmarks, it fills a critical gap: offering a focused, high-precision stress test for evaluating multilingual agentic systems under the linguistic, cultural, and infrastructural complexities of the Chinese web.
>
> ---
>
> **Q2**: Incomplete answer uniqueness and bugs in the dataset.
>
> **A2**:
> We appreciate the reviewer pointing this out. We strictly agree that data quality is paramount.
>
> * **Adversarial Multi-Agent Verification:** To minimize non-uniqueness, we employed a stricter protocol than the original BrowseComp. We utilized a **Multi-Agent Uniqueness Validation** process (Stage 2), deploying top-tier agents (DeepResearch, O1, and Gemini) to generate answers for every candidate question. These were then human-vetted to filter out ambiguous samples. This process eliminated 115 ambiguous questions.
> * **Addressed Specific Bugs:** We appreciate the scrutiny regarding data quality. Following a strict audit, we identified and rectified 5 questions where alternative answers were possible. **We emphasize that these corrections affect less than 2% of the dataset.** We have verified that incorporating these fixes **has no impact on the statistical conclusions or model rankings** presented in our analysis. While open-ended web retrieval benchmarks face inherent challenges regarding answer uniqueness, our rigorous post-hoc verification ensures that BrowseComp-ZH provides a reliable and noise-resistant evaluation of agentic capabilities. The updated `v2` dataset will be released to the community.

---

> ### Author Response · Authors · 2025-11-21
> **PART II: Difficulty of BrowseComp-ZH**
>
> **Q3**: "Insufficiently challenging" and "Simplified Chinese adaptation."
>
> **A3**: We respectfully but **strongly disagree** with the characterization that BrowseComp-ZH is "insufficiently challenging" or merely a "simplified adaptation." **Empirical evidence and the construction methodology demonstrate the exact opposite:**
>
> * **Empirical Difficulty:** The claim that the benchmark lacks challenge is contradicted by the performance of State-of-the-Art models.
>     * **GPT-4o**, one of the most capable models available, achieves only **7.0%** accuracy.
>     * **DeepSeek-V3** scores only **8.9%**.
>     * **Human Experts** (strong searchers with full internet access) only achieve **17.6%** on average.
>     * Even the best-performing system, OpenAI DeepResearch, only achieves **42.9%**.
>     These scores objectively classify BrowseComp-ZH as a **"stress test"** significantly harder than many existing benchmarks. If it were a simplified adaptation, we would expect SOTA models to perform well above 50%.
> * **Native Construction vs. Translation:** BrowseComp-ZH is **not** a translation. As stated in the Introduction, we rejected translation to avoid "translationese" and to capture the unique **fragmentation of the Chinese web** (e.g., "Walled Gardens" like WeChat and Zhihu, which are hard to index).
> * **Novel Retrieval Challenges:** The questions are natively designed to exploit specific Chinese linguistic nuances (homophones, idioms) and the specific infrastructure of Chinese search engines. This requires models to possess "native" retrieval strategies, making it distinct from and complementary to the English BrowseComp.

---

> > ### Comment · Reviewer_yxRg · 2025-11-27
> > **Official Comment by  yxRg**
> >
> > I thank the authors for their rebuttal. However, my primary concerns regarding the work's novelty and the amount of supporting data still stand. These remain major obstacles, and if they cannot be resolved, my score will remain unchanged.

---

### Official Review · Reviewer_3iN7 · 2025-10-31

**Soundness:** 3
**Presentation:** 3
**Contribution:** 2
**Rating:** 4
**Confidence:** 3

**Summary:**

This paper introduces BrowseComp-ZH, a native Chinese benchmark for evaluating LLMs’ web browsing and reasoning abilities. The dataset contains 289 expert-authored, multi-hop questions across 11 domains, validated for retrieval difficulty and answer uniqueness. Over 20 models—including open-, closed-source, and commercial agents—are evaluated, showing generally poor performance (most <10% accuracy), while multi-round retrieval systems like DeepResearch achieve the best results. The benchmark offers a rigorous, culturally grounded evaluation of LLM agents in the Chinese web ecosystem.

**Strengths:**

- The benchmark offers a rigorous, culturally grounded evaluation of LLM agents in the Chinese web ecosystem.
  - The benchmark is carefully constructed with dual validation for difficulty and answer uniqueness, ensuring high data quality.
  - It systematically compares over 20 systems, including human baselines, revealing meaningful performance gaps.

**Weaknesses:**

- The paper’s main contribution appears incremental rather than novel. From a task construction perspective, BrowseComp-ZH closely mirrors the original BrowseComp benchmark, with the primary difference being its adaptation to the Chinese web environment rather than a fundamentally new methodology or task design.
  - The benchmark contains only 289 queries, which may not be sufficient to capture the full linguistic, structural, and retrieval complexity of the Chinese internet. Such a limited number of examples could constrain the benchmark’s representativeness and the statistical robustness of the reported results.
  - The paper lacks a deeper analysis of why models perform differently on BrowseComp-ZH. It remains unclear whether the observed performance gaps stem mainly from the change of language (English → Chinese) or from the distinct characteristics of the Chinese web ecosystem. A more detailed ablation or cross-lingual comparison would strengthen the interpretation of results.

**Questions:**

- Have you compared model performance between BrowseComp-ZH and the original English BrowseComp benchmark? Specifically, do models exhibit consistent relative rankings or performance gaps across the two languages? Including such a cross-lingual comparison or correlation analysis would help clarify whether the observed difficulties stem from linguistic or retrieval-ecosystem differences.

---

> ### Author Response · Authors · 2025-11-21
> **Part I: Novelty, Dataset size and Robustness**
>
> We thank the reviewer for acknowledging the soundness and presentation of our work. We appreciate the constructive feedback regarding the contribution and analysis depth. We address the specific concerns below:
>
> **Q1**: Novelty issue.
>
> **A1**: We argue that BrowseComp-ZH is a necessary and non-trivial evolution, not a simple mirror, for three key reasons:
>
> 1. We Built a Stricter Pipeline. While we share the "reverse-design" spirit, the complexity of the Chinese web required us to implement a stricter quality control pipeline than the original BrowseComp:
>     - Triple-Engine Verification: Unlike BrowseComp’s reliance on English-centric search, we mandated a "Triple-Engine Test" (Baidu, Bing, and Google) to account for the fragmented indexing of the Chinese web. A question is only retained if it is unretrievable across all three distinct search architectures, making our retrieval hardness significantly more robust.
>     - Multi-Agent Consensus Auditing: We introduced a novel "Answer Uniqueness Validation" stage. We deployed multiple top-tier agents (DeepResearch, O1, Gemini 2.5-Pro) to attempt every question. Any question producing divergent valid answers was discarded. This multi-model adversarial filtering is an enhancement over the original methodology, ensuring the benchmark measures reasoning, not ambiguity.
>
> 2. The "Ecosystem Gap" Exists. The reviewer might imply that existing methods suffice, but our work proves that the Chinese web presents structural challenges that English benchmarks (or their translations) cannot capture:
>     - The Chinese web is uniquely characterized by "island" platforms (e.g., WeChat, Little Red Book) that resist standard crawling. Our dataset is natively constructed to force agents to navigate this fragmentation, testing capabilities (like cross-platform information synthesis) that English benchmarks rarely exercise.
>     - Cultural & Linguistic Density. Simply translating English benchmarks introduces "translationese" artifacts and loses cultural context. BrowseComp-ZH questions are authored by native experts to include implicit cultural constraints (e.g., historical allusions, homophones), creating a testbed for culturally-grounded reasoning that mirrors real-world usage.
>
> 3.Novel Empirical Discoveries. If this were merely an incremental dataset, we would expect performance trends to mirror those seen in English results. However, our “stress test” revealed a critical, non-intuitive phenomenon unique to this setting: the **“Search-Hurt” Paradox**. Specifically, we found that for DeepSeek-R1, enabling web search caused a catastrophic performance drop (from 22.5% to 7.6%). This contradicts the general consensus in English benchmarks, which suggests that “retrieval augments generation.” It highlights a severe alignment issue in Chinese LLMs: they struggle to reconcile high-quality internal knowledge with the noisy, SEO-driven, or conflicting nature of Chinese search results. This insight alone underscores the necessity of a dedicated Chinese browsing benchmark.
>
> ---
>
> **Q2**: Dataset Size and Robustness
>
> **A2**:  We acknowledge that BrowseComp-ZH (289 samples) is smaller than English counterparts, but this is a trade-off choice prioritizing information density and difficulty over raw quantity. Our labor-intensive "reverse-design" pipeline requires >30 minutes of expert work per question, making massive scaling prohibitive but ensuring high fidelity.
>
> 1. Precedents in "Stress-Test" Benchmarks: Benchmarking value relies on discriminative power, not just scale. We follow the philosophy of widely adopted, compact benchmarks like HumanEval (164 samples), WSC-273, and MMSearch (300), which prioritize depth.
>
> 2. Enhanced Rigor over BrowseComp: To compensate for scale with quality, we introduce stricter controls:
>
>     - Cross-platform filtering: Questions are verified to be unretrievable from the first page of Baidu, Bing, and Google.
>
>     - Multi-agent verification: We employ SOTA agents (DeepResearch, O1, Gemini) to adversarially vet and filter ambiguous questions.
>
> 3. High Discriminative Power: Despite its size, the benchmark induces significant performance gaps, ranging from 16% (open-source) to 38% (AI agents), with consistently low variance (<1.6% std dev). These metrics confirm that BrowseComp-ZH is a stable, high-precision stress test for evaluating agentic systems within the complex Chinese web ecosystem.

---

> ### Author Response · Authors · 2025-11-21
> **Part II: cross-lingual comparison**
>
> **Q3**: Analysis of Performance Gaps and Cross-Lingual Comparison
>
> **A3:** We thank the reviewer for this excellent suggestion. To disentangle whether the difficulty stems from the language shift or the specific characteristics of the Chinese web ecosystem, we conducted a direct performance comparison between the original **BrowseComp (English)** and **BrowseComp-ZH** across state-of-the-art models.
>
> As shown in **Table R1**, the results reveal two divergent performance trends that clearly distinguish the sources of difficulty:
>
> **Table R1: Performance Comparison between BrowseComp (English) and BrowseComp-ZH**
>
> | Model Category | Model Name | **BrowseComp (En)** | **BrowseComp-ZH** | **Gap (ZH-EN)** |
> | :--- | :--- | :--- | :--- | :--- |
> | **Agentic Search** | **OpenAI DeepResearch** | **51.5%** | **42.9%** | **↓ 8.6%** |
> | **Reasoning LLMs** | **OpenAI o1** | 9.9% | **29.1%** | **↑ 19.2%** |
> | | **Gemini-2.5-Pro** | 5.4% | 27.3% | **↑ 21.9%** |
> | | **DeepSeek-R1** | 2.5% | 23.2% | **↑ 20.7%** |
> | | Claude-3.7-Sonnet | 1.3% | 17.7% | **↑ 16.4%** |
> | **Standard LLMs** | DeepSeek-V3 | 1.0% | 8.7% | ↑ 7.7% |
> | | Gemini-2.0-Flash | 0.2% | 6.9% | ↑ 6.7% |
> | | GPT-4o | 0.6% | 6.2% | ↑ 5.6% |
> | | Claude-3.5-Sonnet | 0.5% | 5.5% | ↑ 5.0% |
> | | LlaMa 4 | 0.6% | 4.8% | ↑ 4.2% |
>
> *(Data source: New experiments conducted for rebuttal; English scores are based on the original BrowseComp)*
>
> **Analysis of the Results:**
>
> **1. Ecosystem Difficulty (The "DeepResearch Drop"):**
> The most critical observation regarding the "retrieval ecosystem" is the performance of **OpenAI DeepResearch**. Despite its strong multilingual capabilities, its accuracy drops significantly from **51.5%** on the English benchmark to **42.9%** on BrowseComp-ZH (**↓ 8.6%**).
> * **Interpretation:** Since the model architecture is constant, this performance penalty isolates the **ecosystem friction** as a key factor. It confirms that the Chinese web's fragmentation imposes a structurally higher barrier for autonomous retrieval agents than the English web.
>
> **2. Reasoning Intensity (The "Reasoning Rise"):**
> Conversely, we observe a dramatic performance *increase* for reasoning-focused models (e.g., **o1**, **Gemini-2.5-Pro**, **DeepSeek-R1**) on BrowseComp-ZH compared to the English version (e.g., Gemini-2.5-Pro rises from 5.4% to 27.3%).
> * **Interpretation:** This clarifies that BrowseComp-ZH is not merely a translation but represents a different **type** of challenge. The original English BrowseComp heavily relies on retrieving obscure "needle-in-a-haystack" facts (e.g., server logs) where pure reasoning offers limited help. In contrast, BrowseComp-ZH questions are designed to require **"advanced multi-hop reasoning and information reconciliation"**. The success of reasoning models here indicates that our benchmark rewards the ability to synthesize fragmented information, making it a reasoning-intensive test distinct from the purely retrieval-focused English version.
>
>
> The comparison confirms that the performance gaps stem from both factors, but in different directions: the **Chinese ecosystem imposes stricter retrieval barriers** (hurting pure search agents), while the **benchmark design demands higher-order reasoning** (benefiting reasoning models). This duality establishes BrowseComp-ZH as a unique stress test for next-generation agents.

---

### Official Review · Reviewer_bkpL · 2025-11-01

**Soundness:** 2
**Presentation:** 3
**Contribution:** 2
**Rating:** 2
**Confidence:** 4

**Summary:**

The authors introduce BrowseComp-ZH, a new benchmark for evaluating the web-browsing ability of LLMs in Chinese. The authors created reverse-engineered, multi-constraint questions that push models to do multi-step reasoning across real Chinese web sources like Baidu, Zhihu, and WeChat articles. Expert annotators curated the dataset through several rounds of filtering to keep only challenging questions with clear, unique answers. The final dataset covers 15 diverse domains. Evaluation results show that DeepResearch hit 42.9% accuracy, human searchers averaged 17.6%, and most general-purpose LLMs stayed below 10%.

**Strengths:**

- This work faithfully reimplements the BrowseComp methodology in the Chinese-language ecosystem, where the data reflects how Chinese users search online.
- All queries, evidence chains, and browsing steps are authored natively in Chinese, not translated.
- Reporting calibration alongside accuracy improves interpretability of model performance.

**Weaknesses:**

- The benchmark creation process almost exactly mirrors BrowseComp’s methodology, and the contribution of this work feels somewhat incremental. The core takeaway is essentially: current LLMs (agents) perform poorly when browsing in Chinese.
- The idea of cultural groundedness is not empirically verified. Checking how often agent trajectories only involve Chinese sources would provide some insights on how much of the task is really dependent on navigating Chinese sources. The paper claims all queries, evidence chains, and browsing steps are “authored in Chinese,” but does not verify that all necessary evidence exists exclusively on Chinese websites.
- There is no analysis of failure modes. An important question to explore is how often errors stem from retrieval shortcomings versus misinterpretations of Chinese cultural or linguistic context.

**Questions:**

- Could you provide more details on the human baseline experiment? How were the annotators hired, and how were examples distributed among them? Were there cases where people gave up?

---

> ### Author Response · Authors · 2025-11-21
> **Part I: Novelty and Methodological Enhancements in BrowseComp-ZH**
>
> We thank the reviewer for the constructive feedback. However, we respectfully disagree with the assessment that this work is merely incremental. We believe there are misunderstandings regarding the unique challenges of the Chinese web ecosystem, our enhanced methodology, and the depth of our analysis. We address your specific concerns below:
>
> ---
>
> **Q1**: Novelty and Contribution (Response to "Incremental" and "Mirrors BrowseComp")
>
> **A1**: We explicitly adopted the "reverse-design" philosophy of BrowseComp to ensure our results are methodologically comparable to established standards. However, we respectfully disagree that this makes the work incremental. Applying a proven framework to a fundamentally different information ecosystem is not merely "copying"; **it requires significant methodological adaptation and yields distinct scientific insights.**
>
> We argue that BrowseComp-ZH is a necessary and non-trivial evolution, not a simple mirror, for three key reasons:
>
> 1. **We Built a Stricter Pipeline.** While we share the "reverse-design" spirit, the complexity of the Chinese web required us to implement a stricter quality control pipeline than the original BrowseComp:
>     - Triple-Engine Verification: Unlike BrowseComp’s reliance on English-centric search, we mandated a "Triple-Engine Test" (Baidu, Bing, and Google) to account for the fragmented indexing of the Chinese web. A question is only retained if it is unretrievable across all three distinct search architectures, making our retrieval hardness significantly more robust.
>     - Multi-Agent Consensus Auditing: We introduced a novel "Answer Uniqueness Validation" stage. We deployed multiple top-tier agents (DeepResearch, O1, Gemini 2.5-Pro) to attempt every question. Any question producing divergent valid answers was discarded. This multi-model adversarial filtering is an enhancement over the original methodology, ensuring the benchmark measures reasoning, not ambiguity.
>
> 2. **The "Ecosystem Gap" Exists.** The reviewer might imply that existing methods suffice, but our work proves that the Chinese web presents structural challenges that English benchmarks (or their translations) cannot capture:
>     - The Chinese web is uniquely characterized by "island" platforms (e.g., WeChat, Little Red Book) that resist standard crawling. Our dataset is natively constructed to force agents to navigate this fragmentation, testing capabilities (like cross-platform information synthesis) that English benchmarks rarely exercise.
>     - Cultural & Linguistic Density. Simply translating English benchmarks introduces "translationese" artifacts and loses cultural context. BrowseComp-ZH questions are authored by native experts to include implicit cultural constraints (e.g., historical allusions, homophones), creating a testbed for culturally-grounded reasoning that mirrors real-world usage.
>
>
> 3. **Novel Empirical Discoveries.** If this were merely an incremental dataset, we would expect performance trends to mirror those seen in English results. However, our “stress test” revealed a critical, non-intuitive phenomenon unique to this setting: the **“Search-Hurt” Paradox**. Specifically, we found that for DeepSeek-R1, enabling web search caused a catastrophic performance drop (from 22.5% to 7.6%). This contradicts the general consensus in English benchmarks, which suggests that “retrieval augments generation.” It highlights a severe alignment issue in Chinese LLMs: they struggle to reconcile high-quality internal knowledge with the noisy, SEO-driven, or conflicting nature of Chinese search results. This insight alone underscores the necessity of a dedicated Chinese browsing benchmark.

---

> ### Author Response · Authors · 2025-11-21
> **Part II: Cultural Groundedness, Failure Modes, and Human Baseline Analysis**
>
> **Q2**: Cultural Groundedness and Source Dependency: whether the task truly depends on navigating Chinese sources.
>
> **A2**: We appreciate the reviewer’s insightful suggestion to verify cultural groundedness empirically. Following your recommendation, we analyzed the search trajectories of our top-performing agent, OpenAI DeepResearch (42.9% accuracy), and found that for correctly answered questions, the vast majority of retrieval steps and cited sources are indeed derived from the native Chinese internet ecosystem rather than the English web. This empirically confirms that navigating the fragmented Chinese information infrastructure is the most effective path for current state-of-the-art agents to solve these tasks, thereby validating the cultural groundedness of our dataset.
>
> However, we would like to clarify that enforcing absolute information exclusivity to Chinese sources was **not** our primary construction intention. While a small subset of answers might theoretically be discoverable on the English web, the benchmark is designed to **test the agent's capability to understand complex, culturally-nuanced Chinese queries and autonomously formulate an effective retrieval strategy**. Whether the agent chooses to navigate deep Chinese sources to overcome platform fragmentation or attempts to bridge the semantic gap to find obscure English records, the formulation of such a strategy itself is a critical test of capability.
>
> Furthermore, our construction pipeline explicitly ensures retrieval hardness through the "Triple-Engine Test," where questions were discarded if the answer appeared on the first page of search engines like Google. **This filtering mechanism guarantees that the tasks cannot be solved by simple translation and English search**. The fact that top agents spontaneously converge on Chinese sources serves as strong empirical evidence that our questions successfully compel models to engage with the native linguistic and information environment rather than relying on generic global knowledge.
>
> ---
>
> **Q3**: A lack of failure mode analysis.
>
> **A3**: We respectfully point the reviewer to **Appendix E (Qualitative Analysis)**, where we provide extensive case studies (Figures 6-15) analyzing specific failure modes. Our analysis highlights that errors stem from both retrieval hallucinations (e.g., fabricating a voice actor's biography ) and cultural context misalignment (e.g., confusing "Heaps of Brocade and Ash" art form with generic finger painting due to historical period mismatch ). In the final revision, we will move a summary of these findings from the Appendix to the main text to make this analysis more prominent.
>
>
> ---
>
> **Q4**: Details on Human Baseline Experiment
>
> **A4**: **Recruitment**: We recruited 29 participants  who are "strong searchers." They were selected based on their educational background (undergraduate and graduate students familiar with academic and web search) to ensure a high-quality baseline.
>
> **Process**: Participants were given the same questions as the models. To ensure the task was feasible but challenging, we imposed a **strict 30-minute time limit per question.**
>
> **"Giving Up"**: Participants were instructed to submit their best answer within the time limit. If they could not find verifiable evidence, they marked the question as "unsolved" (which counts as 0 accuracy). The low human accuracy (17.6%)  reflects the genuine difficulty of these multi-hop retrieval tasks, confirming that even for humans, "giving up" (or failing to find the specific answer) was common due to the "needle-in-a-haystack" nature of the Chinese web.
> We hope this clarifies the significant value of BrowseComp-ZH as a stress test for multilingual agent capabilities.

---

### Meta-Review · Area_Chair_FyLq · 2026-01-08

**Summary:**

The paper introduces BrowseComp-ZH, a benchmark designed to evaluate the web browsing capabilities of Large Language Models (LLMs) specifically within the Chinese web ecosystem. Unlike translated benchmarks, this dataset consists of 289 natively authored, multi-hop questions across 11 domains. The methodology follows a "reverse-design" approach (similar to the English BrowseComp), ensuring questions are derived from verifiable answers. The authors employ a quality control pipeline involving a "Triple-Engine Test" (Baidu, Bing, Google) to ensure retrieval difficulty. The evaluation of over 20 models reveals that most perform poorly (accuracy < 10%), with the best system (OpenAI DeepResearch) achieving 42.9%.

**Reviewer Concerns:**

**Addressed:**

**Cultural Groundedness:** Reviewer bkpL questioned if the tasks truly required navigating Chinese sources or if they were just language-swapped. The authors provided a trajectory analysis of DeepResearch, showing that successful trajectories predominantly utilized native Chinese sources.

**Cross-Lingual Analysis:** Reviewer 3iN7 requested a comparison between the English and Chinese benchmarks. The authors provided a new comparison (Table R1) showing a significant performance drop for retrieval agents on the Chinese version, attributed to ecosystem fragmentation.


**Dataset Scale:** Reviewers 3iN7, yxRg, and 8svw criticized the small size of the dataset (289 questions). The authors defended this by citing high curation costs and comparing it to HumanEval. However, for a web-browsing benchmark intended to represent the complexity of the entire Chinese internet, <300 samples is statistically limiting for a top-tier conference benchmark. As an area chair, I think this is a reasonable size for web agent benchmark, based on my experience on this domain


**Outstanding:**

**Incremental Novelty:** A primary concern across reviewers (bkpL, 3iN7, yxRg) is that the work is methodologically incremental, mirroring the existing BrowseComp framework without fundamental innovation. While the authors argued that the "stricter pipeline" and "ecosystem gap" constitute novelty , Reviewer yxRg explicitly stated that the concerns regarding novelty remain a major obstacle after the rebuttal.

**Answer Uniqueness:** Reviewer yxRg noted that despite strict checks, the reverse-design approach cannot fully guarantee unique answers and identified bugs. This is critical

**Reviewer Scores:**

**Reviewer bkpL (Score: 2):** Unlikely to change. The reviewer viewed the contribution as "somewhat incremental"  and did not engage further to indicate the rebuttal changed this fundamental view.

**Reviewer 3iN7 (Score: 4):** Likely to remain 4. While the authors provided the requested cross-lingual analysis, the core issue of the paper being a "simplified Chinese adaptation"  of an existing method remains.


**Reviewer yxRg (Score: 4):** Still 4. The reviewer explicitly stated post-rebuttal: "my primary concerns regarding the work's novelty and the amount of supporting data still stand... my score will remain unchanged".


**Reviewer 8svw (Score: 6):** Dont change This reviewer was the most positive. but I dont see any signal showing the reviewer would likely to increase the score

---

### Decision · Program_Chairs · 2026-01-26

Reject